# KGAREVION: AN AI AGENT FOR KNOWLEDGE-INTENSIVE BIOMEDICAL QA

**Xiaorui Su**[1]    **Yibo Wang**[2]    **Shanghua Gao**[1]    **Xiaolong Liu**[2]    **Valentina Giunchiglia**[3]
**Djork-Arné Clevert**[4]    **Marinka Zitnik**[1]
[1]Harvard University    [2]University of Illinois Chicago    [3]Imperial College London    [4] Pfizer
`xiaorui_su@hms.harvard.edu, ywang633@uic.edu,`
`shanghua_gao@hms.harvard.edu, xliu262@uic.edu,`
`v.giunchiglia20@imperial.ac.uk,`
`Djork-Arne.Clevert@pfizer.com, marinka@hms.harvard.edu`

## ABSTRACT

Biomedical reasoning integrates structured, codified knowledge with tacit, experience-driven insights. Depending on the context, quantity, and nature of available evidence, researchers and clinicians use diverse strategies, including rule-based, prototype-based, and case-based reasoning. Effective medical AI models must handle this complexity while ensuring reliability and adaptability. We introduce KGAREVION, a knowledge graph-based agent that answers knowledge-intensive questions. Upon receiving a query, KGAREVION generates relevant triplets by leveraging the latent knowledge embedded in a large language model. It then verifies these triplets against a grounded knowledge graph, filtering out errors and retaining only accurate, contextually relevant information for the final answer. This multi-step process strengthens reasoning, adapts to different models of medical inference, and outperforms retrieval-augmented generation-based approaches that lack effective verification mechanisms. Evaluations on medical QA benchmarks show that KGAREVION improves accuracy by over 5.2% over 15 models in handling complex medical queries. To further assess its effectiveness, we curated three new medical QA datasets with varying levels of semantic complexity, where KGAREVION improved accuracy by 10.4%. The agent integrates with different LLMs and biomedical knowledge graphs for broad applicability across knowledge-intensive tasks. We evaluated KGAREVION on AfriMed-QA, a newly introduced dataset focused on African healthcare, demonstrating its strong zero-shot generalization to underrepresented medical contexts.

## 1 INTRODUCTION

Biomedical reasoning requires integrating diagnostic and therapeutic decision-making with an understanding of the biology and chemistry of the disease of drugs (Patel et al., 2005). Large language models (LLMs)(OpenAI, 2024; Dubey et al., 2024; Gao et al., 2024) demonstrate strong general capabilities, but their responses to medical questions often suffer from incorrect retrieval, omission of key information, and misalignment with current scientific and clinical knowledge. These models can struggle to generate contextually relevant answers that account for local factors, such as patient demographics, geographic variations, and specialized biomedical domains(Harris, 2023). A limitation arises from their inability to integrate multiple types of evidence by combining *structured, codified* scientific knowledge with *tacit, noncodified* expertise—clinical intuition, case-based experience, and learned heuristics, which are essential for contextualizing scientific evidence within real-world medical decision-making (Harris, 2023).

LLM-powered QA models often lack the *multi-source* and *grounded knowledge* required for effective medical reasoning. The answer to complex medical questions requires an understanding of the intricate and highly specialized nature of biomedical concepts. However, LLMs trained in general-purpose data struggle to solve medical problems that require *specialized knowledge in the domain*. This challenge stems from their inability to differentiate subtle, domain-specific nuances of medical

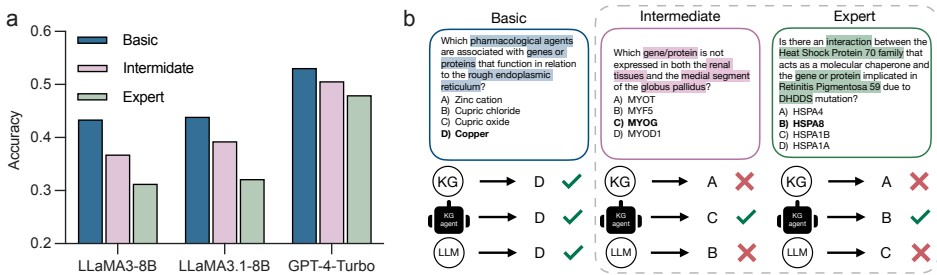

Figure 1: **a)** Performance of existing LLMs on three new datasets (MedDDx-Basic, MedDDx-Intermediate, MedDDx-Expert) introduced in this paper with questions of varying difficulty. **b)** Sample questions from the new datasets.

significance. As a result, LLMs often do not reason effectively in complex scenarios, where successful inference requires recognizing and reasoning about dependencies between multiple interrelated medical concepts within a single question and interpreting highly similar yet semantically distinct biomedical entities with precision, as illustrated in Fig. 1.

To address these limitations, researchers have turned to retrieval-augmented generation (RAG), which follows a retrieve-then-answer paradigm (Shi et al., 2024). These methods enrich LLMs with external knowledge sources, allowing them to retrieve information from biomedical databases and structured repositories (Fan et al., 2024). However, the accuracy of RAG-based answers is heavily dependent on the quality of the retrieved content, making them prone to errors (Karpukhin et al., 2020). Many biomedical knowledge bases contain incomplete, outdated, or incorrect information, leading to unreliable retrieval Adlakha et al. (2024); Thakur et al. (2023). Furthermore, approaches based on RAG lack post-retrieval verification mechanisms, leaving them unable to assess whether retrieved information is factually accurate, contextually relevant, or if it omits essential information (Zhao et al., 2023).

Knowledge graphs (KG) of medical concepts serve as grounded knowledge bases that provide precise and specialized in-domain knowledge for medical QA models (Qi et al., 2024; Murali et al., 2023; Chandak et al., 2023). Although KGs improve these models, they often contain gaps, limiting their ability to capture biomedical relationships. Approaches that retrieve medical concepts based solely on direct associations (edges) in a KG fail to account for implicit or complex relationships. For example, two proteins with different biological functions may lack a direct connection in a KG, even though they share biological similarity at the molecular level (Menche et al., 2015). Advancing LLM-powered models for knowledge-intensive medical QA requires systems that simultaneously capture complex associations among multiple medical concepts, integrate multi-source knowledge systematically, and verify retrieved information to ensure contextual accuracy and relevance.

**Present work.** We introduce KGAREVION, a KG-based LLM agent designed for complex biomedical QA that integrates the non-codified knowledge of LLMs with the structured, codified knowledge found in KGs. As illustrated in Fig. 2, KGAREVION executes four key actions to ensure accurate and context-aware biomedical reasoning. First, it prompts the LLM to generate relevant triplets based on the input question. To effectively leverage structured KG data, KGAREVION fine-tunes the LLM on a KG completion task, incorporating pre-trained structural embeddings of triplets as prefix tokens. The fine-tuned model then evaluates the correctness of the generated triplets. Next, KGAREVION performs a 'Revise' action to correct erroneous triplets, refining the knowledge base before selecting the final answer. Given the complexity of medical reasoning, KGAREVION adaptively chooses the most appropriate reasoning approach for each question, allowing for nuanced and context-aware QA. This flexibility enables KGAREVION to handle both *multi-choice* and *open-ended* QA tasks.

Our key contributions include: ① Developing KGAREVION, a versatile KG agent that dynamically adjusts reasoning strategies, achieving a 6.75% improvement over 15 baseline models in seven datasets, including three challenging newly curated benchmarks. ② Demonstrating that grounding through generated triplets significantly enhances KGAREVION's capabilities across multiple KGs. ③ Showing that KGAREVION effectively answers complex, knowledge-intensive medical queries in both multi-choice and open-ended QA formats. ④ Evaluating KGAREVION on African healthcare datasets: We benchmark KGAREVION on AfriMed-QA, a newly introduced

dataset focused on African healthcare. The results highlight KGAREVION's strong zero-shot generalization, demonstrating its ability to reason effectively in underrepresented medical contexts. ⑤ Analyzing robustness to input variations: We analyze KGAREVION's sensitivity to changes in question structure, answer ordering, and answer relabeling. Unlike LLMs, which exhibit high variance when answer choices are reordered, KGAREVION maintains stable performance, demonstrating its stronger robustness in real-world settings. KGAREVION is available at https://github.com/mims-harvard/KGARevion.

## 2 RELATED WORK

**LLM-based reasoning.** General-purpose LLMs (GPT (OpenAI, 2024), LLaMA family (Dubey et al., 2024; Touvron et al., 2023), Mistral (Jiang et al., 2023)) and LLMs fine-tuned on biomedical data (BioMedLM (Venigalla et al., 2022), Codex (Liévin et al., 2024), MedAlpaca (Han et al., 2023), Med-PaLM (Singhal et al., 2023), PMC-LLaMA (Wu et al., 2024a)) leverage their vast embedded knowledge to perform medical reasoning. Some models enhance reasoning by decomposing complex queries into sub-tasks, solving them step by step using structured prompts, as seen in Chain-of-Thought (CoT) (Wei et al., 2024) and CODEX CoT (Gramopadhye et al., 2024). However, these methods struggle with knowledge-intensive medical queries that require multi-source domain-specific knowledge, leading to gaps in accuracy and completeness.

**RAG-based models.** Self-RAG (Asai et al., 2024) is a framework that enhances LLM performance through retrieval and self-reflection. LLM-AMT (Wang et al., 2023b) improves medical question answering by integrating authoritative medical textbooks into LLMs with specialized knowledge retrieval and self-refinement techniques. Adaptive-RAG (Jeong et al., 2024) introduces a dynamic RAG framework that adapts retrieval strategies based on the complexity of the questions. However, its performance is restricted by the quality of the knowledge retrieved (Zhang et al., 2024).

**KG-based models.** Models, such as QAGNN (Yasunaga et al., 2021), JointLK (Sun et al., 2022), and Dragon (Yasunaga et al., 2022), handle medical questions solely using KGs in an end-to-end manner. However, these methods cannot be easily applied to questions involving unseen nodes or incomplete knowledge within the graphs. In addition, structured KGs have driven research toward graph-based RAG models, motivating models such as GraphRAG (Edge et al., 2024), KG-RAG (Soman et al., 2023), and MedGraphRAG (Wu et al., 2024b). KG-Rank (Yang et al., 2024) ranks the retrieved triplets and filters out irrelevant knowledge to improve the accuracy of the search. Additionally, GenGround (Shi et al., 2024) uses a Generate-then-Ground pipeline that grounds answers by prompting LLMs to validate retrieved knowledge. However, these approaches rely heavily on semantic dependencies, overlooking the rich structural information within the KGs.

## 3 KGAREVION AGENT

Given a set of biomedical questions $Q$, each question consists of a question stem $q$ and a set of candidate answers $C$. For example, in Fig. 2a, the sample question has the stem $q$ = *"Which gene interacts with the Heat Shock Protein 70 family that acts as a molecular chaperone and is implicated in Retinitis Pigmentosa 59 due to DHDDS mutation?"* along with a set of semantically related candidate answers $C = \{HSPA4, HSPA8, HSPA1B, HSPA1A\}$. The goal is to identify the correct answer using both an LLM (denoted as $P$) and a knowledge graph (KG, denoted as $G$). Here, a KG is represented as a set of triplets $G = \{(h, r, t)\}$, where each triplet consists of a head entity $(h)$, a relationship $(r)$, and a tail entity $(t)$. Table F.2 provides a summary of the notation. We consider both multiple-choice and open-ended reasoning settings (see Results).

KGAREVION is an LLM-powered agent (Wu et al., 2023; Li et al., 2023) that defines agentic actions (Schick et al., 2023; Shen et al., 2023; Nakano et al., 2021) to collaboratively perform complex tasks (Tang et al., 2023; Bran et al., 2023; Boiko et al., 2023). Fig. 2 provides an overview of KGAREVION, which operates through four key actions: Generate (§3.1; Generates triplets relevant to the input question), Review (§3.2; Assesses the correctness of each generated triplet), Revise (§3.3; Corrects any triplet identified as incorrect), and Answer (§3.3; Produces the final answer based on the triplets verified by the Review action). This structured reasoning process allows KGAREVION to integrate both LLM-generated knowledge and structured KG-based validation, improving accuracy and robustness in knowledge-intensive medical QA.

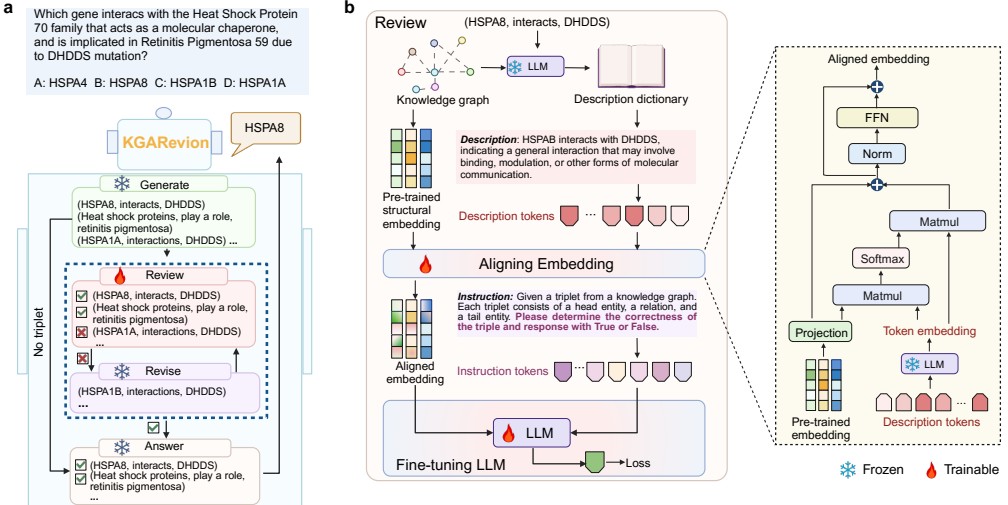

Figure 2: **a)** Overview of KGAREVION agent. **b)** Overview of fine-tuning in the Review action.

## 3.1 GENERATE ACTION

The Generate action aims to gather comprehensive structured knowledge from input questions. Specifically, this action first identifies all medical concepts involved in the input question stem $q$ and then generates a set of triplets $T$ related to the question based on the extracted medical concepts.

Depending on the content of the answer candidate, the input questions can be broadly categorized into two types: choice-aware and non-choice-aware. The answer candidates in the choice-aware group have specific contents, whereas the ones in the non-choice-aware group only contain yes-or-no options (as shown in Appendix Table 4). These different types of questions require distinct reasoning processes: choice-aware questions involve analyzing the content of each answer candidate, while non-choice-aware questions only require focusing on the question stem.

To handle this, this action is designed to prompt the LLM (Ouyang et al., 2022; Wang et al., 2023a) to follow different procedures for generating relevant triplets according to the input question type.

- For choice-aware questions, the Generate action generates triplets based on the contents of each answer candidate and extracted medical concepts in question stem $q$;
- For non-choice-aware questions, the Generation action directly generates triplets based on medical concepts presented in question stem $q$.

The rationale behind this design is that LLMs have inherent biases in their knowledge, often generating more detailed information on familiar topics compared to less familiar ones when all answer candidates are presented simultaneously (Dai et al., 2024). Additionally, this approach helps reduce the impact of the order in which the answer candidates are presented. The process of the Generate action can be formulated as:

$$T = \begin{cases} \{P(q, a_i)\}, & 1 \leq i \leq |C|, \quad \text{if } C \nsubseteq \{\text{Yes}, \text{No}, \text{Maybe}\} \\ P(q), & \text{if } C \subseteq \{\text{Yes}, \text{No}, \text{Maybe}\} \end{cases} \quad (1)$$

where $a_i$ denotes the candidate answer in $C$, and the LLM $P(\cdot)$ is prompted to extracts triplets from the medical concepts involved in its input.

## 3.2 REVIEW ACTION

To enable LLMs to accurately judge the correctness of generated triplets, beyond relying solely on semantic dependencies inferred by LLMs (Shinn et al., 2023), the Review action also leverages the connections and relationships among various medical concepts contained in KGs. This is achieved by fine-tuning the LLM on a KG completion task, explicitly integrating entity embeddings learned from KGs into the LLM. Then the Review action is performed by the fine-tuned LLM to assess the correctness of triplets generated by the Generate action, as shown in Fig. 2.

### 3.2.1 FINE-TUNING STAGE

**Generating KG Embeddings and Triplet Descriptions.** We use the well-known KG representation learning method, TransE (Bordes et al., 2013), to learn structural embeddings for both entities and relations in $G$. For a triplet $(h, r, t) \in G$, the learned pre-trained embeddings are denoted as $\mathbf{e}_h \in \mathbb{R}^d$, $\mathbf{e}_r \in \mathbb{R}^d$, and $\mathbf{e}_t \in \mathbb{R}^d$, where $d$ represents the embedding dimension. These embeddings are kept fixed during LLM fine-tuning. In addition, we instruct the LLM to generate a description template for each relation $r \in G$. We store these descriptions in a dictionary $D(\cdot)$, which can be found in Appendix Table 8.

**Aligning Embedding.** Since the embeddings in LLMs are based on token vocabularies (Radford et al., 2019), LLMs cannot directly interpret pre-trained structural embeddings, as they lack semantic meaning. To make use of the pre-trained structural embeddings, we align them with the corresponding descriptions to generate new embeddings for the input triplet.

Specifically, given the description $D(r)$ for the input triplet $(h, r, t)$, we denote the embedding of $D(r)$ obtained from the LLM as $\mathbf{X} \in \mathbb{R}^{|l| \times d_L}$, where $|l|$ is the maximum number of tokens and $d_L$ is the embedding dimension in the LLM. Next, we concatenate the embeddings of the head entity, the relation, and the tail entity, denoted as $\mathbf{V} = [g(\mathbf{e}_h); g(\mathbf{e}_r); g(\mathbf{e}_t)] \in \mathbb{R}^{3 \times d_L}$, where $g(\cdot) : \mathbb{R}^d \to \mathbb{R}^{d_L}$. We then apply an attention block (Vaswani, 2017), followed by a two-layer feed-forward network (FFN, as shown in Fig. 2b) to obtain the aligned triplet embedding matrix $\mathbf{Z} \in \mathbb{R}^{3 \times d_L}$ as follows:

$$\widehat{\mathbf{V}} = \mathbf{V} + \sigma(\mathbf{V}\mathbf{X}^T)\mathbf{X} \tag{2}$$

$$\mathbf{Z} = \widehat{\mathbf{V}} + ((\varphi(\widehat{\mathbf{V}})\mathbf{W}_1))\mathbf{W}_2 \tag{3}$$

where $\sigma(\cdot)$ is the Softmax function, $\varphi(\cdot)$ represents layer normalization, $\mathbf{W}_1 \in \mathbb{R}^{d_L \times d_h}$ and $\mathbf{W}_2 \in \mathbb{R}^{d_h \times d_L}$ are trainable parameters in the FFN, and $d_h$ is the dimension of the hidden layer in the FFN.

**Fine-tuning LLM.** After obtaining the aligned embedding $\mathbf{Z}$, we add it to the beginning of the instruction and fine-tune the LLM using LoRA (Hu et al., 2022) with the next-token prediction loss (Radford, 2018). The instruction is: *'Given a triple from a knowledge graph. Each triple consists of a head entity, a relation, and a tail entity. Please determine the correctness of the triple and response with True or False.'* The output should be either True or False.

### 3.2.2 INFERENCE STAGE

The fine-tuned LLM is then integrated to the Review action to check the accuracy of each triplet in $T$, which was generated by the Generate action (3.1). Specifically, we first use UMLS codes (Bodenreider, 2004) to map entities in the KG and obtain pre-trained structural embeddings for the head entity, relation, and tail entity, respectively. These embeddings, along with their descriptions and instructions, are fed into the fine-tuned LLM to determine whether the triplet is correct or not.

However, not all entities in the generated triplet $(h, r, t) \in T$ can be mapped to entities in KGs. To address this, the Review action applies a soft constraint rule to distinguish whether the generated triplet is factually wrong or the result of incomplete knowledge in KGs, as follows:

- Factually Wrong: if we can map $h$ and $t$ to entities in KGs and the output of fine-tuned LLM is False, then the triplet $(h, r, t)$ is factually wrong and is removed from $T$.
- Incomplete Knowledge: if we cannot map either $h$ or $t$ to entities in KGs, then the triplet $(h, r, t)$ is considered incomplete knowledge and is kept.

In this way, the triplet in $T$ can be grouped into two categories, i.e., the True triplet set $V$ and False triplet set $F$, where $T = V \cup F$ and $V \cap F = \emptyset$.

### 3.3 REVISE AND ANSWER ACTIONS

If $F$ has triplets, KGAREVION calls the Revise action to adjust the triplets in $F$ to include more triplets covering more medical concepts that help with the answering of the input question. The new generated revised triplets are then reviewed by the Review action to make sure that they are correct and related to the input question. If the Review action outputs "True", then the revised triplets are added to the set of True triplets $V$. Otherwise, KGAREVION continues to call the Revise action until the max round $k$ ($k \geq 1$) is achieved.

After obtaining the set of True triplets from the Review or Revise actions, KGAREVION finally calls the Answer action to prompt the LLM to generate the answer to input question $q$ based on the verified triplets in $V$.

## 4 RESULTS

**Datasets.** We first start with four multi-choice medical QA benchmarks (Xiong et al., 2024a) (Table 1). In addition, we introduce a new benchmark for multi-choice complex medical QA focused on differential diagnosis (DDx), named MedDDx. We begin by collecting questions and corresponding answers from STaRK-Prime (Wu et al., 2024c). For each question, we then select the top three entities with the highest semantic similarity to serve as additional answer candidates. MedDDx comprises a total of 1,769 multi-choice QA samples. Based on the standard deviation of semantic similarity between answer candidates and the correct answer, we categorize the dataset into three difficulty levels: MedDDx-Basic, MedDDx-Intermediate, and MedDDx-Expert (The samples in each dataset are shown in Fig. 1, and details are available in Appendix 5).

**Baselines.** We consider eight LLM-based reasoning models, four RAG-based models, and three KG-based models. The LLM-based reasoning models include LLaMA (2-7B/13B, 3-8B, 3.1-8B) (Touvron et al., 2023; Dubey et al., 2024), Mistral (Jiang et al., 2023), MedAlpaca (7B) (Han et al., 2023), PMC-LLaMA (7B) (Wu et al., 2024a), and LLaMA3-OpenBioLLM-8B (Ankit Pal, 2024). The RAG-based models include Self-RAG (Asai et al., 2024), MedRAG (Xiong et al., 2024b), KG-RAG (Soman et al., 2023), and KG-Rank (Yang et al., 2024). The KG-based models include QAGNN

Table 1: QA benchmarks and three new Med-DDx datasets. 'OR' indicates whether the open-ended reasoning evaluation is done.

| Datasets | Size | QA | OR |
|---|---|---|---|
| MMLU-Med | 1,089 | A/B/C/D | ✔ |
| MedQA-US | 1,273 | A/B/C/D | ✔ |
| PubMedQA* | 500 | Yes/No/Maybe | ✔ |
| BioASQ-Y/N | 618 | Yes/No | ✔ |
| MedDDx-Basic | 483 | A/B/C/D | ✔ |
| MedDDx-Intermediate | 1,041 | A/B/C/D | ✔ |
| MedDDx-Expert | 245 | A/B/C/D | ✔ |

(Yasunaga et al., 2021), JointLK (Sun et al., 2022), and Dragon (Yasunaga et al., 2022).

**Evaluation setup.** We consider two evaluation settings. **Multi-choice reasoning:** This setting evaluates the model's performances on all collected multi-choice QA datasets. The model is tasked to select the correct answer to a user input question from a set of candidate answers. **Open-ended reasoning:** All candidate answers are masked, meaning that the model has to generate a response to the input question independently without being presented with a set of candidate answers. The model produces an answer solely on its own generated response. Additionally, we design two new evaluation scenarios for each setting to test model abilities in solving complex medical questions by considering the number of medical concepts and the semantic similarity among answer candidates. **Query complexity scenario (QSS):** This is a hard evaluation scenario to test how the model performs with the increase of the number of medical concepts present in a question since the question often becomes more intricate, requiring more nuanced inferences between concepts to achieve the correct answer, with the increase of medical concepts. **Semantic complexity scenario (CSS):** This is a harder evaluation scenario that tests the model's ability to identify the correct answer among semantically similar and closely medically related candidate answers.

### 4.1 MULTI-CHOICE QUESTION-ANSWERING

Table 2 shows the accuracy and variance of KGAREVION and all baselines on all datasets. Evaluation on four gold standard medical QA datasets shows that KGAREVION improves the average accuracy by over 4.8%, outperforming all baselines in handling medical queries.

**Results under QSS.** Fig. 3a illustrates the trends of KGAREVION alongside the top-performing baselines as the number of medical concepts increases. It can first be observed that KGAREVION outperforms baselines of the same size, regardless of the number of medical concepts involved. Moreover, KGAREVION maintains stable performance as the number of medical concepts increases and even improves when processing questions with n = 6, compared to those with n = 5. In contrast, baseline models struggle with complex questions containing 5 or 6 medical concepts. Such an

Table 2: The accuracy of KGAREVION and all baselines on four gold standard and three newly created datasets under multi-choice reasoning settings. The value highlighted in Blue indicates the best result among LLM-based reasoning models with a size smaller than 8B, including LLaMA3-8B, while Red marks the top value among LLaMA3.1-8B and other types of baselines. std means the standard deviation under three runs.

| Multi-choice Reasoning | Established Medical QA Benchmarks | | | | MedDDx | | |
|---|---|---|---|---|---|---|---|
| Method | MMLU-Med | MedQA-US | PubMedQA* | BioASQ-Y/N | Basic | Intermediate | Expert |
| Metrics | Acc. (std) | Acc. (std) | Acc. (std) | Acc. (std) | Acc. (std) | Acc. (std) | Acc. (std) |
| LLaMA2-7B | 0.376 (.006) | 0.281 (.004) | 0.448 (.010) | 0.568 (.006) | 0.215 (.030) | 0.198 (.004) | 0.192 (.012) |
| LLaMA2-7B (CoT) | 0.318 (.005) | 0.251 (.002) | 0.465 (.011) | 0.547 (.011) | 0.289 (.010) | 0.265 (.006) | 0.229 (.023) |
| Mistral-7B | 0.634 (.004) | 0.477 (.007) | 0.400 (.002) | 0.644 (.001) | 0.412 (.003) | 0.356 (.003) | 0.375 (.007) |
| Mistral-7B (CoT) | 0.634 (.003) | 0.474 (.002) | 0.372 (.005) | 0.651 (.002) | 0.404 (.010) | 0.368 (.023) | **0.379** (.027) |
| MedAlpaca-7B | 0.600 (.004) | 0.401 (.001) | 0.333 (.015) | 0.493 (.034) | 0.399 (.012) | 0.325 (.004) | 0.311 (.009) |
| MedAlpaca-7B (CoT) | 0.603 (.004) | 0.399 (.003) | 0.315 (.015) | 0.485 (.025) | 0.395 (.007) | 0.321 (.011) | 0.312 (.010) |
| PMC-LLaMA-7B | 0.207 (.011) | 0.247 (.004) | 0.179 (.007) | 0.346 (.017) | 0.087 (.015) | 0.086 (.002) | 0.079 (.006) |
| PMC-LLaMA-7B (CoT) | 0.204 (.008) | 0.208 (.002) | 0.125 (.014) | 0.208 (.002) | 0.088 (.002) | 0.077 (.004) | 0.063 (.005) |
| LLaMA3-8B | 0.634 (.005) | **0.566** (.004) | **0.586** (.008) | 0.654 (.005) | 0.428 (.015) | 0.319 (.002) | 0.306 (.009) |
| LLaMA3-8B (CoT) | **0.651** (.003) | 0.552 (.003) | 0.574 (.002) | **0.681** (.002) | **0.434** (.010) | **0.368** (.004) | 0.313 (.003) |
| Llama3-OpenBioLLM-8B | 0.636 (.005) | 0.383 (.003) | 0.350 (.026) | 0.623 (.005) | 0.238 (.004) | 0.235 (.011) | 0.229 (.020) |
| Llama3-OpenBioLLM-8B (CoT) | 0.571 (.003) | 0.295 (.002) | 0.283 (.001) | 0.646 (.004) | 0.370 (.013) | 0.330 (.001) | 0.327 (.011) |
| LLaMA3.1-8B | 0.677 (.007) | **0.563** (.006) | 0.596 (.009) | 0.687 (.006) | 0.434 (.018) | 0.368 (.002) | 0.306 (.021) |
| LLaMA3.1-8B (CoT) | **0.681** (.005) | 0.549 (.003) | **0.600** (.005) | 0.706 (.002) | **0.439** (.005) | 0.393 (.005) | 0.322 (.014) |
| LLaMA2-13B | 0.442 (.002) | 0.253 (.004) | 0.252 (.004) | 0.455 (.002) | 0.286 (.003) | 0.338 (.006) | 0.317 (.006) |
| LLaMA2-13B (CoT) | 0.415 (.002) | 0.354 (.005) | 0.232 (.006) | 0.422 (.003) | 0.309 (.005) | 0.263 (.013) | 0.243 (.016) |
| QAGNN | 0.317 (.003) | 0.450 (.005) | 0.439 (.033) | 0.644 (.002) | 0.295 (.003) | 0.265 (.002) | 0.253 (.003) |
| JointLK | 0.288 (.005) | 0.472 (.003) | 0.468 (.007) | 0.640 (.007) | 0.247 (.004) | 0.250 (.003) | 0.244 (.004) |
| Dragon | 0.319 (.003) | 0.475 (.002) | 0.472 (.005) | 0.646 (.003) | 0.286 (.003) | 0.247 (.005) | 0.240 (.004) |
| Self-RAG (7B) | 0.322 (.019) | 0.380 (.028) | 0.534 (.028) | 0.594 (.012) | 0.238 (.007) | 0.199 (.037) | 0.224 (.045) |
| Self-RAG (13B) | 0.502 (.004) | 0.408 (.020) | 0.331 (.158) | 0.646 (.050) | 0.249 (.010) | 0.290 (.018) | 0.266 (.031) |
| KG-Rank (13B) | 0.452 (.005) | 0.362 (.011) | 0.305 (.019) | 0.503 (.015) | 0.253 (.021) | 0.256 (.013) | 0.234 (.010) |
| KG-RAG (8B) | 0.516 (.005) | 0.343 (.001) | 0.429 (.017) | 0.662 (.005) | 0.434 (.021) | **0.413** (.007) | **0.391** (.004) |
| MedRAG (70B) | 0.579 (.004) | 0.487 (.014) | 0.574 (.022) | **0.719** (.018) | 0.365 (.004) | 0.348 (.011) | 0.327 (.003) |
| KGAREVION (LLaMA3, w/o Review) | 0.621 (.002) | 0.528 (.003) | 0.556 (.002) | 0.713 (.004) | 0.310 (.006) | 0.334 (.004) | 0.313 (.008) |
| KGAREVION (LLaMA3, w/o Revise) | 0.657 (.004) | 0.594 (.006) | 0.562 (.002) | 0.723 (.005) | 0.386 (.008) | 0.372 (.004) | 0.327 (.003) |
| KGAREVION (LLaMA3, $k=1$) | **0.703** (.004) | 0.610 (.010) | 0.562 (.002) | **0.744** (.003) | **0.473** (.004) | 0.404 (.006) | 0.395 (.003) |
| KGAREVION (LLaMA3, $k=2$) | 0.696 (.006) | 0.616 (.008) | 0.566 (.002) | 0.723 (.010) | 0.457 (.006) | 0.414 (.006) | 0.395 (.005) |
| KGAREVION (LLaMA3, $k=3$) | 0.678 (.006) | **0.628** (.002) | **0.590** (.005) | 0.737 (.007) | 0.469 (.008) | **0.451** (.004) | **0.411** (.005) |
| **Improvement over best baseline** | **+5.2%** | **+6.2%** | **+0.4%** | **+6.3%** | **+3.9%** | **+8.3%** | **+3.2%** |
| KGAREVION (LLaMA3.1, w/o Review) | 0.695 (.006) | 0.546 (.002) | 0.560 (.004) | 0.736 (.003) | 0.298 (.015) | 0.299 (.003) | 0.327 (.006) |
| KGAREVION (LLaMA3.1, w/o Revise) | 0.716 (.006) | 0.573 (.005) | 0.568 (.011) | 0.749 (.003) | 0.392 (.012) | 0.337 (.006) | 0.352 (.005) |
| KGAREVION (LLaMA3.1, $k=1$) | **0.734** (.004) | 0.618 (.002) | 0.619 (.004) | **0.763** (.001) | **0.483** (.013) | **0.457** (.010) | 0.409 (.005) |
| KGAREVION (LLaMA3.1, $k=2$) | 0.720 (.003) | 0.616 (.005) | 0.656 (.006) | 0.745 (.005) | 0.396 (.003) | 0.454 (.008) | 0.342 (.004) |
| KGAREVION (LLaMA3.1, $k=3$) | 0.716 (.004) | **0.620** (.003) | **0.638** (.004) | 0.749 (.003) | 0.469 (.012) | 0.411 (.010) | **0.447** (.005) |
| **Improvement over best baseline** | **+5.3%** | **+5.7%** | **+3.8%** | **+4.4%** | **+4.4%** | **+4.4%** | **+5.6%** |

observation indicates that KGAREVION advances in handling complex medical questions involving multiple medical concepts.

**Results under CSS.** As seen in Table 2, evaluations on three difficult levels in MedDDx indicate that KGAREVION exhibits a strong ability in handling differential diagnosis questions that request professional and accurate knowledge. In addition, the obtained results also show that KGAREVION excels in identifying the correct answer among semantically similar answer candidates since it improves the accuracy on MedDDx-Expert by 3.2% and 5.6% with LLaMA3-8B and LLaMA3.1-8B as the backbone LLM, respectively.

## 4.2 OPEN-ENDED REASONING

We transform multiple-choice questions into descriptive, open-ended ones to better simulate real-world medical scenarios, where such inquiries are more common (see details in Appendix B.4). This adjustment requires our model to generate responses without predefined choices, encouraging holistic reasoning and the integration of diverse knowledge sources. By removing answer choices, we can more effectively assess the reasoning ability of KGAREVION in complex medical situations, resulting in a more realistic evaluation of its capabilities. Table 3 shows the accuracy and variance across all datasets. The variance here denotes the difference in accuracy compared with that in the multi-choice reasoning setting.

**Results under QSS.** Fig. 3b shows the accuracy obtained by pure LLM and KGAREVION with the increase of medical concepts under open-ended reasoning setting. Compared to pure LLMs in the open-ended reasoning setting, KGAREVION shows a significant improvement in handling complex medical reasoning tasks involving more than 4 medical concepts and a comparable performance in questions with less than 3 medical concepts.

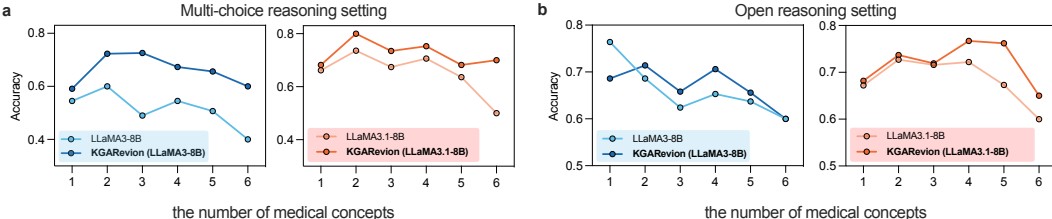

Figure 3: The accuracy of KGAREVION and pure LLMs with the medical concepts increase under **a**) multi-choice reasoning setting and **b**) open-ended reasoning setting.

Table 3: The accuracy of KGAREVION and baselines on four gold standard and three newly created datasets under open-ended reasoning settings. The value highlighted in Blue indicates the best result among LLM-based reasoning models with a size smaller than 8B, including LLaMA3-8B, while Red marks the top value among LLaMA3.1-8B and other types of baselines. ΔAcc. denotes the difference in performance between the open-ended and multiple-choice reasoning settings.

| Open-ended Reasoning | Open-ended inquiries without pre-defined choices | | | | MedDDx-No-Opt | | (Open-ended) |
|---|---|---|---|---|---|---|---|
| Method | MMLU-Med | MedQA-US | PubMedQA* | BioASQ-Y/N | Basic | Intermediate | Expert |
| Metrics | Acc. ΔAcc. | Acc. ΔAcc. | Acc. ΔAcc. | Acc. ΔAcc. | Acc. ΔAcc. | Acc. ΔAcc. | Acc. ΔAcc. |
| LLaMA2-7B | 0.328 (-0.048) | 0.302 (+0.021) | 0.546 (+0.098) | 0.625 (+0.057) | 0.286 (+0.071) | 0.305 (+0.107) | 0.302 (+0.110) |
| LLaMA2-7B (CoT) | 0.362 (+0.044) | 0.243 (-0.008) | 0.418 (-0.047) | 0.642 (+0.095) | 0.265 (-0.024) | 0.270 (+0.005) | 0.280 (+0.051) |
| Mistral-7B | 0.591 (-0.043) | 0.412 (-0.065) | 0.344 (-0.056) | 0.629 (-0.015) | 0.249 (-0.163) | 0.228 (-0.128) | 0.273 (-0.102) |
| Mistral-7B (CoT) | 0.583 (-0.051) | 0.398 (-0.076) | 0.212 (-0.160) | 0.657 (+0.006) | 0.245 (-0.159) | 0.232 (-0.136) | 0.286 (-0.093) |
| PMC-LLaMA-7B | 0.073 (-0.134) | 0.082 (-0.165) | 0.090 (-0.089) | 0.139 (-0.201) | 0.249 (+0.162) | 0.191 (+0.105) | 0.232 (+0.153) |
| PMC-LLaMA-7B (CoT) | 0.080 (-0.124) | 0.079 (-0.129) | 0.092 (-0.033) | 0.125 (-0.083) | 0.220 (+0.132) | 0.245 (+0.168) | 0.228 (+0.165) |
| LLaMA3-8B | 0.595 (-0.039) | 0.458 (-0.108) | 0.532 (-0.054) | 0.672 (+0.018) | 0.343 (-0.085) | 0.314 (-0.005) | 0.317 (+0.011) |
| LLaMA3-8B (CoT) | 0.608 (-0.043) | 0.449 (-0.103) | 0.562 (-0.012) | 0.714 (+0.033) | 0.289 (-0.145) | 0.304 (-0.064) | 0.327 (+0.014) |
| Llama3-OpenBioLLM-8B | 0.324 (-0.312) | 0.157 (-0.226) | 0.157 (-0.193) | 0.324 (-0.299) | 0.016 (-0.222) | 0.006 (-0.229) | 0.008 (-0.221) |
| Llama3-OpenBioLLM-8B (CoT) | 0.398 (-0.173) | 0.146 (-0.149) | 0.100 (-0.183) | 0.142 (-0.539) | 0.098 (-0.272) | 0.084 (-0.246) | 0.108 (-0.219) |
| LLaMA3.1-8B | 0.607 (-0.070) | 0.551 (-0.012) | 0.514 (-0.082) | 0.694 (-0.007) | 0.322 (-0.112) | 0.289 (-0.079) | 0.335 (+0.029) |
| LLaMA3.1-8B (CoT) | 0.697 (+0.016) | 0.563 (+0.014) | 0.572 (-0.028) | 0.706 (-0.000) | 0.306 (-0.133) | 0.294 (-0.099) | 0.315 (-0.007) |
| LLaMA2-13B | 0.348 (-0.094) | 0.283 (+0.030) | 0.218 (-0.034) | 0.421 (-0.034) | 0.190 (-0.096) | 0.123 (-0.215) | 0.153 (-0.164) |
| LLaMA2-13B (CoT) | 0.311 (-0.104) | 0.267 (-0.087) | 0.266 (+0.034) | 0.471 (+0.049) | 0.269 (-0.040) | 0.268 (+0.005) | 0.282 (+0.039) |
| Self-RAG (7B) | 0.256 (-0.066) | 0.235 (-0.145) | 0.316 (-0.218) | 0.379 (-0.215) | 0.167 (-0.071) | 0.246 (+0.047) | 0.213 (-0.011) |
| Self-RAG (13B) | 0.309 (-0.193) | 0.297 (-0.111) | 0.438 (+0.107) | 0.539 (-0.107) | 0.212 (-0.037) | 0.232 (-0.058) | 0.226 (-0.040) |
| KG-Rank (13B) | 0.151 (-0.301) | 0.189 (-0.173) | 0.203 (-0.102) | 0.188 (-0.315) | 0.127 (-0.126) | 0.133 (-0.123) | 0.133 (-0.101) |
| KG-RAG (8B) | 0.310 (-0.206) | 0.290 (-0.053) | 0.316 (-0.113) | 0.359 (-0.303) | 0.216 (-0.218) | 0.220 (-0.193) | 0.213 (-0.178) |
| KGAREVION (LLaMA3, w/o Review) | 0.645 (+0.024) | 0.609 (+0.081) | 0.552 (-0.004) | 0.701 (-0.012) | 0.400 (+0.090) | 0.360 (+0.026) | 0.356 (+0.043) |
| KGAREVION (LLaMA3, w/o Revise) | 0.668 (+0.011) | 0.626 (+0.032) | 0.572 (+0.010) | 0.716 (-0.007) | 0.426 (+0.040) | 0.403 (+0.031) | 0.412 (+0.085) |
| KGAREVION (LLaMA3, $k = 1$) | 0.687 (-0.016) | 0.628 (+0.018) | 0.578 (+0.016) | 0.730 (-0.014) | 0.465 (-0.008) | 0.430 (+0.026) | 0.428 (+0.033) |
| KGAREVION (LLaMA3, $k = 2$) | 0.682 (-0.014) | 0.638 (+0.022) | 0.566 (-0.000) | 0.736 (+0.013) | 0.527 (+0.070) | 0.463 (+0.049) | 0.489 (+0.094) |
| KGAREVION (LLaMA3, $k = 3$) | 0.696 (+0.018) | 0.632 (+0.004) | 0.572 (-0.018) | 0.733 (-0.004) | 0.489 (+0.020) | 0.411 (-0.040) | 0.429 (+0.018) |
| **Improvement over best baseline** | **+8.8%** | **+18.0%** | **+1.6%** | **+2.2%** | **+18.4%** | **+14.9%** | **+16.2%** |
| KGAREVION (LLaMA3.1, w/o Review) | 0.659 (-0.036) | 0.526 (-0.020) | 0.556 (-0.004) | 0.726 (-0.010) | 0.457 (+0.159) | 0.435 (+0.136) | 0.439 (+0.112) |
| KGAREVION (LLaMA3.1, w/o Revise) | 0.695 (-0.021) | 0.626 (+0.053) | 0.556 (-0.012) | 0.736 (-0.013) | 0.489 (+0.097) | 0.436 (+0.099) | 0.451 (+0.099) |
| KGAREVION (LLaMA3.1, $k = 1$) | 0.720 (-0.014) | 0.644 (+0.026) | 0.560 (-0.059) | 0.757 (-0.006) | 0.469 (-0.014) | 0.454 (-0.003) | 0.437 (+0.028) |
| KGAREVION (LLaMA3.1, $k = 2$) | 0.704 (-0.016) | 0.636 (+0.020) | 0.572 (-0.084) | 0.734 (-0.011) | 0.469 (+0.073) | 0.446 (-0.008) | 0.432 (+0.090) |
| KGAREVION (LLaMA3.1, $k = 3$) | 0.712 (-0.004) | 0.639 (+0.019) | 0.562 (-0.076) | 0.748 (-0.001) | 0.449 (-0.020) | 0.470 (+0.059) | 0.451 (+0.004) |
| **Improvement over best baseline** | **+2.4%** | **+8.1%** | **+0.0%** | **+4.9%** | **+16.7%** | **+17.6%** | **+11.6%** |

**Results under CSS.** To evaluate the model's ability to solve complex medical QA with differential diagnosis, we compare KGAREVION and all baselines on newly created datasets, as shown in Table 3. KGAREVION still achieves the best performance in the open-ended reasoning setting. In addition, compared with the results in multi-choice reasoning setting, KGAREVION performs better. On the one hand, such a result demonstrates the strong ability of KGAREVION in the open-ended reasoning setting. On the other hand, it also indicates that these semantically candidates may affect the reasoning process to a certain extent.

## 4.3 ABLATION ANALYSES

**Effect of the 'Review' action.** As shown in Tables 2 and 3 and in Fig. 4 (KGAREVION (w/o Review) vs. KGAREVION (w/o Revise)), the Review action plays a crucial role in answering medical questions across both reasoning settings. It improves the average accuracy by 9% in the multiple-choice setting and by 4% in the open-ended setting. Fig. 4 shows that the Review action has a more pronounced effect on the MedDDx dataset than the other four datasets in both reasoning settings. This suggests that incorporating the Review action enhances KGAREVION's performance on complex medical questions. Furthermore, the Review action leads to greater accuracy improvements in the gold-standard datasets under the open-ended reasoning setting than in the multiple-choice setting. These findings emphasize the importance of verifying candidate answers for open-ended reasoning, where retrieval and knowledge validation are critical for accuracy.

**Effect of refinement rounds in the 'Revise' action.** The Revise action enhances accuracy by iteratively correcting erroneous triplets until the Review action verifies them as correct. Tables 2 and 3, along with Fig. 4, demonstrate that KGAREVION benefits from this refinement process across both reasoning settings. Fig. 4 shows that the Revise action significantly improves performance on the MedDDx dataset, yielding accuracy gains of 3.3% and 3.0% in the multiple-choice and open-ended settings,

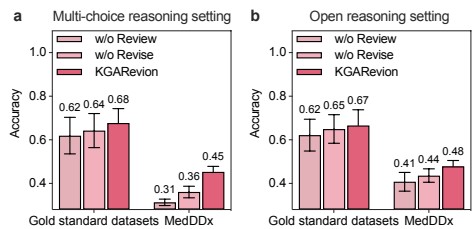

Figure 4: The results of ablation studies across all datasets under two settings.

respectively. We further analyze the impact of the number of revision rounds across all datasets in both settings, as detailed in Tables 2 and 3. The results indicate that KGAREVION achieves optimal performance with $k = 1$ in multiple-choice reasoning datasets. However, it benefits from additional rounds of refinement when handling complex questions, such as those in the MedDDx-Expert dataset. KGAREVION typically requires more iterations to refine and verify the correct answer for open-ended reasoning, highlighting the importance of iterative knowledge validation in knowledge-intensive medical QA.

## 4.4 VERSATILITY OF KGAREVION

**KGAREVION integrates with different LLMs.** KGAREVION is a versatile agent that operates effectively with various LLMs. We implement KGAREVION using three models: LLaMA3-8B, LLaMA3.1-8B, and GPT-4-Turbo. The averaged results across all datasets, shown in Fig. 5a, demonstrate that KGAREVION consistently enhances the performance of its backbone LLMs, improving accuracy by 6% with LLaMA3-8B, 7% with LLaMA3.1-8B, and 2% with GPT-4-Turbo. These improvements highlight the adaptability of KGAREVION across different model architectures, reinforcing its effectiveness in knowledge-intensive QA.

**KGAREVION adapts to different KGs.** The Review action in KGAREVION verifies generated triplets using structured knowledge from KGs. To evaluate how different KGs impact performance (see Appendix C.5 for details), we implement KGAREVION with two KGs and assess its performance across all datasets, as shown in Fig. 5b. The results show that KGAREVION maintains strong performance regardless of the KG used, demonstrating its robustness and generalizability. Despite PrimeKG (Chandak et al., 2023) being

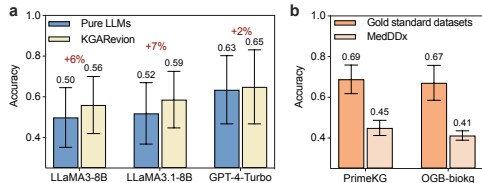

Figure 5: **a)** Performance of KGAREVION with different backbone LLMs across all datasets. **b)** Performance of KGAREVION with different KGs used in the fine-tuning stage of the Review action.

significantly larger than OGB-biokg (Hu et al., 2020), KGAREVION's performance remains high.

This stability arises because KGAREVION uses KGs exclusively in the Review action to validate generated triplets rather than as a primary knowledge retrieval source. Unlike KG-based RAG models, which depend heavily on the structure and completeness of their KGs, KGAREVION only employs KGs to ensure that generated triplets align with medical knowledge. This verification-based approach makes KGAREVION more resilient and less dependent on any single knowledge source, allowing it to outperform RAG models that rely extensively on specific KGs for retrieval.

## 4.5 EVALUATION OF KGAREVION ON AFRICAN HEALTHCARE

To assess whether KGAREVION relies on the extensive pre-trained knowledge of LLMs from existing datasets, we evaluate it on AfriMed-QA, a newly published QA dataset released after all baseline models in this study (Olatunji et al., 2024). We benchmark KGAREVION and five baselines on this dataset under a multiple-choice setting. Further details on AfriMed-QA are provided in Appendix B.3.

Fig. 6 presents the accuracy of KGAREVION compared to the baselines on all expert-level multiple-choice questions in AfriMed-QA. Among the baselines, GPT-4-Turbo achieves the highest accuracy

at 65.0%, while LLaMA3.1-8B reaches 55.5%. To demonstrate KGAREVION's effectiveness, we implement it with both LLaMA3.1-8B and GPT-4-Turbo. The results show that KGAREVION improves accuracy by 5.2% when using LLaMA3.1-8B and by 4.6% when using GPT-4-Turbo. These findings highlight KGAREVION's robustness on a previously unseen dataset and its strong zero-shot generalization capabilities in underrepresented medical contexts.

## 4.6 SENSITIVITY ANALYSES

LLMs often exhibit sensitivity to the ordering and indexing of candidate answers in multiple-choice setups (Zheng et al., 2023; Pezeshkpour & Hruschka, 2023). Prior studies have shown that LLMs are not robust multiple-choice selectors and tend to favor answers appearing in the first position (Li et al., 2024). To examine this issue, we analyze how variations in answer order and indexing impact model performance. We evaluate KGAREVION using LLaMA3-8B and LLaMA3.1-8B as backbones and compare its performance with their LLM-only counterparts across all datasets (detailed results in Appendix C.4.1). Fig. 7 presents the accuracy changes when altering the order or relabeling candidate answers.

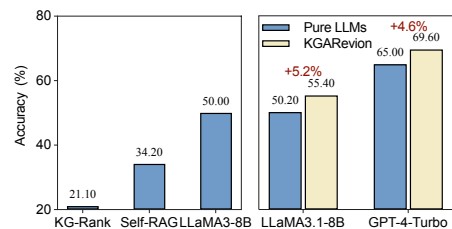

Figure 6: Performance of baselines and KGAREVION with LLaMA3.1-8B and GPT-4-Turbo on AfriMed-QA expert multichoice questions.

**Impact of answer order in multiple-choice setups.** Fig. 7a shows that pure LLMs exhibit strong sensitivity to answer order, with average accuracy shifts of 8.4% for LLaMA3-8B and 16.0% for LLaMA3.1-8B. In contrast, KGARE-VION demonstrates significantly greater robustness, reducing this sensitivity. This stability arises from KGAREVION's evaluations of answers using its Generate action that mitigates the impact of answer order on performance.

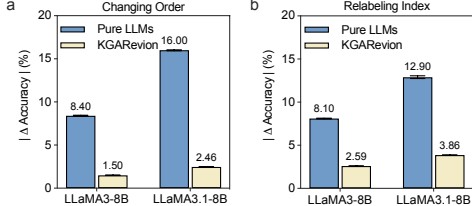

Figure 7: $|\Delta\text{Accuracy}|$ of LLMs and KGAREVION when changing order or relabeling index.

**Impact of answer indexing in multiple-choice setups.** LLMs also show substantial accuracy shifts when relabeling answers from ABCD to EFGH, as illustrated in Fig. 7b. LLaMA3-8B experiences an average accuracy shift of 8.1%, while LLaMA3.1-8B shows a shift of 12.9%. In contrast, KGAREVION significantly stabilizes these LLMs, reducing accuracy loss to 2.59% and 3.86%, respectively. These results highlight KGAREVION's ability to enhance robustness in knowledge-intensive medical reasoning.

## 5 CONCLUSION

Biomedical reasoning requires integrating multi-source, grounded, and specialized domain knowledge to ensure accurate and reliable decision-making. In this work, we introduced KGAREVION, a KG-based LLM agent that addresses these challenges by combining the non-codified knowledge embedded in LLMs with the structured, codified knowledge of medical concepts stored in KGs. By dynamically generating, verifying, and refining knowledge, KGAREVION adapts to the complexity of biomedical QA to generate accurate and contextually relevant answers. Evaluations across multiple-choice and open-ended tasks, including newly curated challenging benchmarks, demonstrate that KGAREVION consistently improves accuracy over existing methods. By grounding LLM-generated knowledge in KGs, KGAREVION enhances reasoning robustness and ensures alignment with biomedical knowledge. These results highlight the potential of KGAREVION to advance knowledge-intensive medical QA, establishing the foundation for future applications in clinical decision support, biomedical research, and complex medical inference.

ACKNOWLEDGMENT

We gratefully acknowledge the support of NIH R01-HD108794, NSF CAREER 2339524, US DoD FA8702-15-D-0001, Harvard Data Science Initiative, Amazon Faculty Research, Google Research Scholar Program, AstraZeneca Research, Roche Alliance with Distinguished Scientists, Sanofi iDEA-iTECH, Pfizer Research, Gates Foundation (INV-079038), Chan Zuckerberg Initiative, John and Virginia Kaneb Fellowship at Harvard Medical School, Biswas Computational Biology Initiative in partnership with the Milken Institute, Harvard Medical School Dean's Innovation Fund for the Use of Artificial Intelligence, and Kempner Institute for the Study of Natural and Artificial Intelligence at Harvard University. V.G. is supported by the UK Medical Research Council, MR/W00710X/1. Any opinions, findings, conclusions or recommendations expressed in this material are those of the authors and do not necessarily reflect the views of the funders.

ETHICS STATEMENT

This study evaluates LLMs on benchmark datasets for medical QA. Although LLMs demonstrate promising capabilities in processing and generating medical information, their use in real-world medical decision-making carries inherent risks. LLMs may produce hallucinations (factually incorrect or misleading information), which can lead to inaccurate medical advice. In addition, these models are trained on datasets that can contain biases, which can lead to disparities in healthcare recommendations. Ensuring transparency, interpretability, and fairness in medical AI systems remains an ongoing challenge. Further, medical LLMs should not be considered a substitute for professional medical advice. The deployment of such models must be approached with caution, ensuring alignment with medical ethics, patient safety guidelines, and regulatory standards. Users must be aware of these limitations, and appropriate human oversight is necessary when applying LLMs in clinical or consumer-facing applications.

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

APPENDIX

## A  EXTENDED RELATED WORK

**Fact-checking Models.** The vast amount of generated text has created a growing need for fact-checking (Graves & Glaisyer, 2012). Early methods relied on manually defined rules to verify generated statements (Hassan et al., 2014; Wu et al., 2012; Jiang et al., 2011). KGs, as structured and reliable knowledge bases, have also been utilized for fact-checking by analyzing graph connectivity. These methods (Tunstall-Pedoe, 2010; Shi & Weninger, 2016; Ciampaglia et al., 2015) validate generated statements by identifying paths between entities mentioned in the statement. Additionally, some approaches use popular KG representation learning algorithms for KG completion tasks to verify the accuracy of statements (Li et al., 2011). With the rise of LLMs, new fact-checking approaches, e.g. RAG-based models, have emerged to ensure the consistency and reliability of generated outputs by integrating external evidence retrieval for verifying and supporting LLM-generated content (Asai et al., 2024).

## B  DATASETS

### B.1  GOLD STANDARD MULTI-CHOICE MEDICAL QA DATASET

In this work, we use four well-known multi-choice medical QA datasets to evaluate the model performance, including two medical examination QA datasets (MMLU-Med, MedQA-US) and two biomedical research QA datasets (PubMedQA*, BioASQ-Y/N). These datasets are derived from (Xiong et al., 2024a). The samples in these datasets are shown in Table 4.

| Dataset Name | Sample |
| --- | --- |
| MMLU-Med | Which of the following best describes the structure that collects urine in the body?

A: Bladder B: Kidney C: Ureter D: Urethra |
| MedQA-US | A microbiologist is studying the emergence of a virulent strain of the virus. After a detailed study of the virus and its life cycle, he proposes a theory: Initially, a host cell is co-infected with 2 viruses from the same virus family. Within the host cell, concomitant production of various genome segments from both viruses occurs. Ultimately, the different genome segments from the viruses are packaged into a unique and novel virus particle. The newly formed virus particle is both stable and viable and is a new strain from the virus family that caused the outbreak of infection. Which of the following viruses is capable of undergoing the above-mentioned process?

A: Epstein-Barr virus B: Human immunodeficiency virus C: Rotavirus D: Vaccinia virus |
| PubMedQA* | Is anorectal endosonography valuable in dyschesia?

A: yes B: no C: maybe |
| BioASQ-Y/N | Can losartan reduce brain atrophy in Alzheimer's disease?

A: yes B: no |

Table 4: Examples of four widely used medical QA datasets

### B.2  MEDDDX

MedDDx is a newly constructed dataset designed to test model performance on semantically complex answers. The motivation behind creating this dataset is twofold:

- While LLMs can perform QA tasks, they often rely heavily on semantic dependencies, making it difficult for them to identify the correct answer among semantically similar answer candidates;

- In real-world medical scenarios, researchers often focus on identifying subtle differences between similar molecules, particularly in treatment or diagnostic settings. For instance, proteins may share similar names but have significantly different structures and functions, making it crucial to distinguish these differences to be able to provide accurate answers (see Table 5 for an example).

| Dataset Name | Sample |
|---|---|
| Basic | Can you recommend medications effective against peptic ulcer disease that also suppress Helicobacter pylori in the stomach?

A: Rebamipide B: Ecabet C: Bendazac D: Nepafenac |
| Intermediate | Can you recommend medications that treat both eosinophilic pneumonia and a parasitic worm infection?

A: Thiabendazole B: Albendazole C: Diethylcarbamazine D: Triclabendazole |
| Expert | Which genes or proteins are expressed exclusively in the pericardium and not in either the dorsal or ventral regions of the thalamus?

A: ADH1A B: ADH1C C: ADH4 D: ADH1B |

Table 5: Examples of four widely used medical QA datasets.

Because of these reasons, we construct MedDDx, a multi-choice medical QA dataset that focuses on answering semantically complex multi-choice QA. These questions are sourced from STaRK-Prime (Wu et al., 2024c), which provides both the questions and their corresponding answers. We extract questions with a single correct answer from the STaRK-Prime testing set and transform them into the multi-choice format. To generate three strong alternative answer candidates, we use semantic similarity to increase the difficulty, selecting the top three entities that have the highest semantic similarity as the correct answer. The semantic embeddings used for this process are derived from `text-embedding-ada-002` model from OpenAI.

The semantic similarity is calculated using cosine similarity. We also compute the standard deviation of semantic similarity between the correct answer and the other three candidates. The density distribution of these values is shown in Fig. 8. Based on this distribution, we divide the queries into three complexity groups using quantile analysis: MedDDx-Expert (0-0.02), MedDDx-Intermediate (0.02-0.04), and MedDDx-Basic (>0.04).

### B.3 AFRIMED-QA

AfriMed-QA creates a novel multispecialty open-source dataset of 15,275 pan-African clinically diverse QA to rigorously evaluate LLMs in African healthcare. The dataset is sourced from over 500 clinical and non-clinical contributors across 16 countries and covers 32 clinical specialties. First, because the dataset is sourced from healthcare systems whose data are not online, it means that none of the multiple-choice and open-ended questions and answers can be in the knowledgebase of the LLM. Because of that, this benchmark has not been leaked into KGARevion's LLM. Furthermore, the knowledge cut-off date for KGARevion's LLM was before AfriMed-QA was released. Second, AfriMed-QA is sourced from clinical, private healthcare systems whereas PrimeKG, STaRK-Prime, and OGB-bioKG are sourced from non-clinical, public, biological data repositories. Furthermore, PrimeKG, STaRK-Prime, and OGB-bioKG were all released several years prior to when the AfriMed-QA benchmark was released. As such, AfriMed-QA has certainly not been leaked into KGARevion's KG.

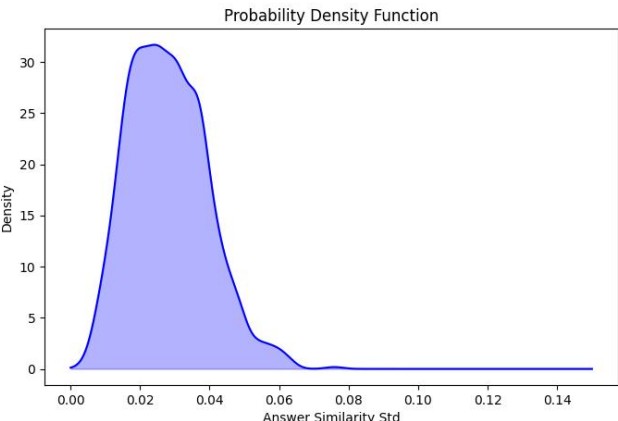

Figure 8: The distribution of the standard deviation of semantic similarities between answer candidates and the correct answer. A lower value indicates greater similarity among the answers.

### B.4 CONVERSION OF MULTI-CHOICE TYPE QUESTIONS TO DESCRIPTIVE TYPE

The conversion of multi-choice questions to descriptive ones is aimed to evaluate real-world medical scenarios where open-ended inquiries are prevalent. To achieve this, we modify the question by adding more descriptive terms, as shown in Table 6.

| Multi-Choice type | Open-ended type |
|---|---|
| Which of the following best describes the structure that collects urine in the body?

A: Bladder B: Kidney C: Ureter D: Urethra | What best describes the structure that collects urine in the body? |
| A microbiologist is studying the emergence of a virulent strain of the virus. After a detailed study of the virus and its life cycle, he proposes a theory: Initially, a host cell is co-infected with 2 viruses from the same virus family. ...... Which of the following viruses is capable of undergoing the above-mentioned process?

A: Epstein-Barr virus B: Human immunodeficiency virus C: Rotavirus D: Vaccinia virus | A microbiologist is studying the emergence of a virulent strain of the virus. After a detailed study of the virus and its life cycle, he proposes a theory: Initially, a host cell is co-infected with 2 viruses from the same virus family. ...... Which virus is capable of undergoing the above-mentioned process? |

Table 6: Examples of conversation of multi-choice type question to descriptive type.

## C IMPLEMENTATION DETAILS

### C.1 EXPERIMENT ENVIRONMENTS

**Hardware.** All experiments are conducted on a machine equipped with 4 NVIDIA H100. We use 1 NVIDIA H100 to implement baselines with small LLMs. In the fine-tuning stage, we use 4 NVIDIA H100 to fine-tune the review module.

**Software.** We implement KGAREVION using Python 3.9.19, PyTorch 2.3.1, Transformers 4.43.1, and Tokenizers 0.19.1. All LLMs adopted in this study are downloaded from Hugging Face, except for OpenAI models.

## C.2 Fine-tuning Details

During the fine-tuning stage, we first split PrimeKG Chandak et al. (2023) into two parts: a training set and a testing set, in a ratio of 8:2. We use LoRA to fine-tune the LLMs on a single machine equipped with 4 NVIDIA H100 GPUs for knowledge graph completion tasks in the Review action.

For hyperparameter tuning, we use grid search to identify the optimal parameter combinations by evaluating the fine-tuned model's performance on the knowledge graph completion task using the testing set. Specifically, we focus on the parameter $r$ in LoRA training and the batch size during the fine-tuning stage. The values explored for $r$ are 16, 32, 64, 128, while the tested batch sizes $bz$ are 128, 256, 512, 1024. The best parameters identified are $r = 32$, $bz = 256$.

## C.3 Implementation Details of Baseline Models

All the results reported in this paper are averaged over multiple runs with different random seeds. For consistency and reproducibility, we select three commonly used seeds: 42, 777, and 1234.

### C.3.1 Implementation Details of Baseline Models under Multi-Choice Setting

**LLM-based Reasoning Models.** We evaluate these LLMs across all datasets in two distinct settings: the traditional inference setting and the CoT inference setting, using the Transformers package (v 4.31.1) with default parameters. The prompts used for evaluation are provided in E.1.

**KG-based Models.** Since the baseline models are trained in an end-to-end way, they cannot be applied directly. To ensure fairness, we train these three models on a collective dataset called MedM-CQA, which contains 4182 question-answering samples, and then evaluate them on all datasets used in this study. In addition, we utilize PrimeKG as the knowledge base for all three methods to construct subgraphs for each sample in both the training and evaluation stages. For knowledge graph processing, we follow the same procedure as JointLK, converting each entity in the KG to its corresponding UMLS code and retrieving entities that match those in the question to construct the subgraph for each question. In the case of Dragon, since it requires a pre-training stage, we first complete its pre-training on the MedMCQA dataset and then directly infer the model on all datasets adopted in this study. We set the epoch number for all three models as 20. All other parameters were derived from the original publications.

**RAG-based Models.** We implement the Self-RAG with Llama2-7B/13B with VLLM (v 0.4.3). We set the temperature in *SamplingParams* as the default value as their original LLMs. MedRAG is implemented using the original code repository. We adopt Llama2-70B as its backbone model, '*textbooks*' as corpus, and MedCPT as retriever. KG-RAG is implemented with the same KG as in KGAREVION, which is PrimeKG and is utilized only during the inference stage. For KG-Rank, we report the results using its original backbone model, which is Llama2-13B and adopt the MedCPT as the ranking method due to limitations with the Cohere API.

### C.3.2 Implementation Details of Baseline Models under Open-ended Reasoning Setting

To test baseline models under the open-ended reasoning setting, we add a new module for both LLM-based reasoning models and RAG-based models. In terms of KG-based models, they could not be deployed under the open-ended reasoning setting due to their model architectures, as shown in E.2.

For each model in the other two groups, we first get a response to the input descriptive question and then adopt the same LLM as their original design to match the response to the content of the correct answer without considering the input question.

## C.4 Implementation Details of Sensitivity Analysis

### C.4.1 Effect of Answer Order/Index

To assess the sensitivity of each model to the answers' order, we swap the positions of two options. For datasets with four options, the order is changed from ABCD to BCAD. For three-option datasets, the order is adjusted from ABC to CAB, and for two-option datasets, it is reversed from AB to BA. It is important to note that we alter the order along with the corresponding content, not just the answer labels. To test the model sensitivity to the answers' index, we relabel the answer indices from ABCD to EFGH. The question obtained after changing the order or the index is presented in Table 7.

|  | Sample |
|---|---|
| Original | Which of the following best describes the structure that collects urine in the body? |
|  | A: Bladder B: Kidney C: Ureter D: Urethra |
| Changing Order | Which of the following best describes the structure that collects urine in the body? |
|  | B: Kidney C: Ureter A: Bladder D: Urethra |
| Relabeling Index | Which of the following best describes the structure that collects urine in the body? |
|  | E: Bladder F: Kidney G: Ureter H: Urethra |

Table 7: Examples of four widely used medical QA datasets.

## C.5 Implementation Details of Adaptability Analysis

To show the flexibility of KGAREVION, we further conduct adaptability analysis by implementing the model with different backbones and KGs. Overall, in this work, we test the KGAREVION with two KGs (i.e., PrimeKG and OGB-biokg). Their details are as follows:

PrimeKG (Chandak et al., 2023) is a precision medicine-oriented knowledge graph that provides a holistic view of diseases. PrimeKG integrates 20 high-quality resources to describe 17,080 diseases with 4,050,249 relationships representing ten major biological scales, including disease-associated protein perturbations, biological processes and pathways, anatomical and phenotypic scale, and the entire range of approved and experimental drugs with their therapeutic action, considerably expanding previous efforts in disease-rooted knowledge graphs.

The OGB-biokg (Hu et al., 2020) dataset contains 5 types of entities: diseases (10,687 nodes), proteins (17,499), drugs (10,533 nodes), side effects (9,969 nodes), and protein functions (45,085 nodes). There are 51 types of directed relations connecting two types of entities, including 38 kinds of drug-drug interactions, 8 kinds of protein-protein interaction, as well as drug-protein, drug-side effect, and function-function relations.

# D KGAREVION

## D.1 Implementation Details in Revise Action

The Revise action is carried out by providing an instruction to an unfine-tuned LLM, such as LLaMA3.1-8B, instead of retraining the model. The instruction directs the LLM to correct the "False" triplet identified by the Review action by modifying either the head or tail entity.

- **Input of False Triplets** The Revise action begins by receiving False triplets, such as (HSPA1A, interactions, DHDDS), which were flagged as False by the Review action.

- **Instruction to the LLM** The system then provides an instruction to the LLM, telling it that the given triplet is incorrect. The instruction asks the LLM to revise the triplet, replacing either the head or tail entity, and to generate a corrected triplet that is relevant to the input question. For example, the instruction could be: *"The following triplet is incorrect:*

*(HSPA1A, interactions, DHDDS). Please revise it to a correct triplet related to the input query."* The details of Instruction is available in E.3

- **Revised Triplet Generation** Simultaneously, the LLM is prompted to output the revised triplet in a specified format, such as a JSON structure: {*'Revised_Triplets': (HSPA1B, interactions, DHDDS)*}.

- **Review of Revised Triplet** Once the revised triplet is generated, KGARevion sends it to the Review action for validation. The Review action checks the correctness of the newly generated triplet.

- **Iterative Revision Process** If the revised triplet is identified as True by the Review action, it is passed on to the Answer action for final output. If the revised triplet is still identified as False, it is sent back to the Revise action for further modification. This process repeats until either a True triplet is identified or the maximum number of revision rounds is reached.

## D.2 IMPLEMENTATION DETAILS IN ANSWER ACTION

The Answer action is performed by prompting an unfine-tuned LLM, such as LLaMA3.1-8B, instead of retraining the model. This approach mirrors the Revise action, allowing the system to leverage pre-trained models without the need for additional training.

- **Input of Verified Triplets** The input to the Answer action consists of all triplets that have been verified as True by the Review actions.

- **Instruction to the LLM** Once the verified triplets are received, KGARevion sends an instruction to the LLM. This instruction directs the LLM to generate the final answer based on the provided triplets.

- **Final Answer Generation** The LLM receives all True triplets and outputs the final answer based on those verified True triplets.

## E PROMPTS

### E.1 PROMPT TEMPLATE FOR EVALUATING BASELINE MODELS UNDER MULTI-CHOICE REASONING SETTING

---
**Prompts for Evaluating LLMs**

The following is a multiple-choice medical question. Please select and provide the correct answer from options 'A', 'B', 'C' or 'D'.

Question: {question}

Answer:

---

---
**Prompts for Evaluating LLMs with CoT**

The following is a multiple-choice medical question. Let's think step by step. Please select and provide the correct answer from options 'A', 'B', 'C' or 'D'.

Question: {question}

Answer:

---

### E.2 PROMPT TEMPLATE FOR EVALUATING BASELINE MODELS UNDER OPEN REASONING SETTING

---

**Prompts for Evaluating LLMs**

The following are medical questions. Please generate a response for input question.

Question: {question}

Answer:

---

---

**Prompts for Evaluating LLMs with CoT**

The following are medical questions. Let's think step by step. Please generate a response for input question.

Question: {question}

Answer:

---

---

**Prompts for Evaluating LLMs**

Given a context, please select the most match answer from options by using 'A', 'B', 'C', and 'D'.

Context: {context}

Options: {options}

Answer:

---

### E.3    PROMPT TEMPLATE IN KGAREVION

The Generate action is implemented with two prompts. One is responsible for identifying medical concepts involved in question stem, the other is for generating triplets.

---

**Prompts for Generate Action**

### Instruction:

Given the following multiple-choice question, extract all relevant medical entities contained within the question stem. Identify and extract all medical entities, such as diseases, proteins, genes, drugs, phenotypes, anatomical regions, treatments, or other relevant medical entities. Ensure that the extracted entities are specific and medically relevant. If no medical entities are found in a particular part, return an empty list for that section. Only return the extracted entities in JSON format with the key "medical_terminologies" and the value is a list of extracted entities.

### Input:

Question: {question}

### Response:

---

---
**Prompts for Generate Action**

### Instruction:

Given the following question stem, medical terminologies, and options, generate a set of related undirected triplets. Each triplet should consist of a head entity, a relation, and a tail entity. The relations should describe meaningful interactions or associations between the entities, particularly in a medical or biomedical context. Use the question stem and the medical entities contained each option to extract triplets that are relevant to the query and can answer the query correctly. Each triplet should be in the format: (Head Entity, Relationship, Tail Entity). Since the triplets are undirected, the order of Head Entity and Tail Entity does not imply any directional relationship between them. The relationship should be one of the following: ['protein_protein', 'carrier', 'enzyme', 'target', 'transporter', 'contraindication', 'indication', 'off-label use', 'synergistic interaction', 'associated with', 'parentchild', 'phenotype absent', 'phenotype present', 'side effect', 'interacts with', 'linked to', 'expression present', 'expression absent']. Ensure that each entity in the triplet is specific and concise, such as diseases, proteins, conditions, symptoms, drugs, treatments, anatomical parts, or other relevant medical entities.

Generate 1-3 triplets for each option, focusing on the ones most relevant to answering the query.

Only return the generated triplets in a structured JSON format with the key as "Triplets" and the value as a list of triplets. The format should be: "Triplets": [(Head Entity, Relationship, Tail Entity), (Head Entity, Relationship, Tail Entity)]

### Input:

Question: {query_stem}

Medical_Terminologies: {medical terminologies}

Options: {option}

### Response:

---

---
**Prompts for Review Action**

Below is an instruction that describes a task, paired with an input that provides further context. Write a response that appropriately completes the request.

### Instruction:

Given a triple from a knowledge graph. Each triple consists of a head entity, a relation, and a tail entity. Taking (PHYHIP, protein_protein, KIF15) as an example, it means that protein PHYHIP has an interaction with protein KIF15. Please determine the correctness of the triple and response True or False. Please directly output 'True' or 'False'.

### Input:

{triplet}

### Response:

---

---

**Prompts for Revise Action**

### Instruction:

Given the following triplet consisting of a head entity, relation, and tail entity, please review and revise the triplet to ensure it is correct and helpful for answering given question. The revision should focus on correcting the head entity, relation, or tail entity as needed to make the triplet accurate and relevant. The triplet should follow the format (head entity, relation, tail entity). Ensure that the revised triplet is factually accurate and contextually appropriate. The relation should clearly define the relationship between the head entity and the tail entity. If no changes are necessary, return the original triplet.

Only return the revised triplet in JSON format with the key 'Revised_Triplets' and the value as the corrected triplet. The format should be: "Revised_Triplets": [(Head Entity, Relationship, Tail Entity)]

### Input:

Triplets: {triplets}

Questions: {query}

### Response:

---

# F    TABLES

## F.1    DESCRIPTION TEMPLATE

| Relation | Description |
|---|---|
| protein_protein | Protein {A} interacts with protein {B}, indicating that the two proteins directly or indirectly associate with each other to perform a biological function. |
| carrier | {A} acts as a carrier for {B}, facilitating its transport or delivery to specific locations within the body or within a cell |
| enzyme | {A} functions as an enzyme that catalyzes a reaction involving {B}, converting it into a different molecule or modifying its structure |
| target | {A} serves as a target for {B}, meaning that {B} binds to or interacts with {A} to exert its biological effect. |
| transporter | {A} is a transporter that facilitates the movement of {B} across cellular membranes or within different compartments of the body. |
| contraindication | The interaction between {A} and {B} is contraindicated, meaning that the presence of one molecule may have adverse effects or reduce the efficacy of the other |
| indication | {A} is indicated for the treatment or management of a condition associated with {B}, suggesting that {A} has a therapeutic role related to {B} |
| off-label use | {A} is used off-label in relation to {B}, meaning it is utilized in a manner not specifically approved but based on clinical judgment. |
| synergistic interaction | {A} and {B} interact synergistically, where their combined effect is greater than the sum of their individual effects |
| associated with | {A} is associated with {B}, indicating a relationship or correlation between the two, often in the context of disease or biological processes |
| parent-child | {A} is related to {B} in a parent-child relationship, where {A} gives rise to or influences the formation of {B} |
| phenotype absent | The interaction between {A} and {B} results in the absence of a specific phenotype, indicating that the normal trait is not expressed |
| phenotype present | The interaction between {A} and {B} results in the presence of a specific phenotype, indicating that a particular trait is expressed |
| side effect | The interaction between {A} and {B} can cause a side effect, where the presence of one molecule leads to unintended and possibly adverse effects |
| interacts with | {A} interacts with {B}, indicating a general interaction that may involve binding, modulation, or other forms of molecular communication. |
| linked to | {A} is linked to {B}, suggesting a connection or association between the two molecules, often in a biological or pathological context. |
| expression present | {A} is expressed in the presence of {B}, indicating that the existence or activity of {B} leads to or correlates with the expression of {A} |
| expression absent | {A} is not expressed in the presence of {B}, indicating that the existence or activity of {B} suppresses or does not correlate with the expression of {A} |

Table 8: The description templates

## F.2 NOTATIONS

| Variable | Description |
|---|---|
| Q | A set of medical queries |
| q | One question stem in Q |
| C | A set of answer candidates for one question stem |
| a | Correct answer in C for the question q |
| P | A large language model |
| G | Knowledge graph |
| h | A head entity in one triplet |
| r | A relationship in one triplet |
| t | A tail entity in one triplet |
| T | A set of triplets generated in the Generation action |
| $a_i$ | One answer candidate in C |
| M | A set of medical concepts in q |
| $\mathbf{e}_h$ | Pre-trained embeddings of h |
| $\mathbf{e}_r$ | Pre-trained embeddings of r |
| $\mathbf{e}_t$ | Pre-trained embeddings of t |
| d | Dimension of pre-trained embeddings |
| f | The fine-tuned model used in Review action |
| b | Bool value that determining the correctness of triplet |
| D | The description dictionary for relationship r |
| $|l|$ | The max length of description tokens |
| $d_L$ | The dimension of token embeddings in LLM |
| $\mathbf{X}$ | Token embedding matrix with the shape of $|l| \times d_p$ obtained from P |
| g | A linear layer to map the dimension of pre-trained embeddings $d$ to that of token embeddings $d_p$ |
| $\mathbf{V}$ | The triplet embedding matrix |
| $\mathbf{Z}$ | Aligned triplet embedding matrix |
| $\sigma(\cdot)$ | The $\mathrm{Softmax}$ function |
| $\varphi(\cdot)$ | The layer normalization function |
| $\mathbf{W}_1$ and $\mathbf{W}_2$ | Trainable parameters in the two layer forward neural network |
| $d_h$ | The dimension of hidden layers in the two layer forward neural network |
| s | An instruction to the LLM |
| V | True triplet set determined by Review action |
| F | False triplet set determined by Review action |
| k | Max round of iterative review and revise actions |
| y | Predicted answer from the Answer action |

Table 9: Additional notation.

