# OpenReview forum: "KGARevion: An AI Agent for Knowledge-Intensive Biomedical QA"
_ICLR.cc/2025/Conference — ICLR 2025 Poster_

### Official Review · Reviewer_pKDu · 2024-10-30

**Soundness:** 4
**Presentation:** 4
**Contribution:** 3
**Rating:** 8
**Confidence:** 3

**Summary:**

This paper introduces KGAREVION, a novel approach that combines the non-codified knowledge of LLMs with the structured knowledge  in KGs to tackle the complexities of knowledge-intensive medical question answering. The method involves extracting relevant triples, using a fine-tuned LLM to verify the correctness of these triples, revising incorrect triples, and generating answers. This approach effectively addresses the challenges in the medical domain that require analogies and the integration of multi-source evidence, demonstrating improved performance across multiple datasets.

**Strengths:**

**Originality**: This paper proposes a new KG-based agent KGAREVION, which can handle complex and knowledge-intensive question answering tasks in the medical field. This work is original because it combines multi-source medical knowledge such as LLMs and KG, and the integration of revise into the processing flow is innovative, especially when dealing with complex problems that require precise domain knowledge.

**Quality**: The quality of the paper is reliable, with each step in the KGAREVION process clearly described and supported by appropriate methodologies. The authors also validate the model's effectiveness through experiments on multiple datasets, including new and challenging ones.

**Clarity**: The paper demonstrates a high level of clarity in presenting the KGAREVION model and its experimental results. The methodology section provides a clear description of the process, with well-organized main diagrams. Additionally, the tables and figures illustrating the experimental results are easy to understand.

**Significance**: This research holds significant importance in the field of medical question answering. It offers an effective tool that integrates data from multiple sources, such as LLMs and KGs, thereby enhancing the ability to accurately address complex medical questions.

**Weaknesses:**

In the description of the methodology, certain parts lack coherence and completeness. For instance, the revise and answer stages do not specify the exact processing methods used.

**Questions:**

**1.** What model is used for the revise process? A more detailed description should be provided. If an unfine-tuned LLM is used, does it have the capability to correct erroneous triples?

**2.** How are triples involving entities not included in the knowledge graph (Incomplete Knowledge) handled? Are they further processed or directly accepted as credible knowledge?

---

> ### Author Response · Authors · 2024-11-16
>
> We sincerely thank the reviewer for their thoughtful and encouraging feedback on the originality, quality, clarity, and significance of KGARevion. We appreciate your positive evaluation of our work and the opportunity to address your insightful comments. In response, we have provided detailed explanations to address your questions. Please do not hesitate to let us know if additional clarifications would be helpful. Thank you again for your valuable input.
>
> ## Weakness 1 Implementation Details of Revise and Answer
> Thank you for your kind suggestion. We have revised the Methods section and the Appendix to include complete information on data processing of the Revise and Answer actions. The updated manuscript now includes an overall description of the method as well as a step-by-step, detailed explanation as follows:
>
> “*The Revise and Answer actions are performed by prompting an LLM, such as Llama3.1-8B, instead of training them. This design ensures that our agent framework remains adaptable and generalizable to various new LLMs without fine-tuning.*
>
> *The Revise action corrects the "False" triplets identified during the Review action. In this step, we prompt the LLM to modify the head or tail entity of each "False" triplet to generate a corrected triplet relevant to the input query. Once the revised triplets are created, the Review action verifies their accuracy. This Revise-Review cycle is repeated until all triplets are verified as correct or a maximum number of iterations is reached.*
>
> *The Answer action uses the verified triplets to generate a response to the query. Specifically, we provide the verified triplets and the query to the LLM, prompting it to produce a final answer based on the verified triplets. Additional details on the prompts and processes for the Revise and Answer actions are included in the Appendix.*”
>
> In the Appendix, we give a detailed description of the methodology for Revise and Answer actions:
>
> ### *Implementation of the Revise Action:*
> *The Revise action is performed by instructing an LLM, such as LLaMA3.1-8B, to correct "False" triplets identified by the Review action. The correction involves modifying either the head or tail entity to generate a valid triplet relevant to the input query.*
>
> + **_Input of False Triplets_**:
>     *The process begins with the Revise action receiving "False" triplets flagged by the Review action. For example, a triplet like (HSPA1A, interacts, DHDDS) may be marked as incorrect.*
> + **_Instruction to the LLM_**:
>     *The system provides clear instructions to the LLM, informing it that the given triplet is incorrect and needs revision. The instruction directs the LLM to replace either the head or tail entity and generate a corrected triplet. For instance: "The following triplet is incorrect: (HSPA1A, interactions, DHDDS). Please revise it to a correct triplet relevant to the input query." Additional details about the instructions are available in the Appendix.*
> + **Revised Triplet Generation**:
>     *The LLM generates a corrected triplet in a specified format, such as a JSON structure. For example: {'Revised_Triplets': (HSPA1B, interactions, DHDDS)}.*
> + **Review of Revised Triplet**:
>     *The revised triplet is then sent to the Review action for validation. The Review action determines whether the newly generated triplet is correct.*
> + **Iterative Revision Process**:
>     *1. If the revised triplet is validated as correct by the Review action, it proceeds to the Answer action for inclusion in the final output.*
>     *2. If the revised triplet is still deemed incorrect, it is sent back to the Revise action for further modification.*
>     *3. This iterative process continues until a valid triplet is identified or the maximum number of revision rounds is reached.*
>
> *This approach ensures that KGARevion generates accurate triplets while minimizing errors through iterative refinement.*
>
>
> ### *Implementation of the Answer Action:*
> *The Answer action is executed by prompting an LLM, such as LLaMA3.1-8B. This approach aligns with the Revise action, enabling the system to leverage pre-trained models without additional training.*
>
> + **_Input of Verified Triplets_**:
> *The Answer action begins with all triplets that have been verified as True by the Review action. These verified triplets serve as the input for generating the final response.*
> + **_Instruction to the LLM_**:
> *KGARevion sends instructions to the LLM, guiding it in generating the final answer using the provided verified triplets.*
> + **_Final Answer Generation_**: *The LLM processes the input of verified triplets and generates a final answer relevant to the input query. This response is based entirely on the correctness and relevance of the verified triplets.*

---

> > ### Author Response · Authors · 2024-11-16
> >
> > ## Q1 Revise Action
> > To ensure that KGARevion (1) remains generalizable to various new LLMs without requiring additional training and (2) allows for fair comparisons across different LLMs, the Generate, Revise, and Answer actions are all performed using a single unfine-tuned LLM, such as LLaMA3-8B or LLaMA3.1-8B. A detailed explanation of these actions is provided in the [response to Weakness 1] and the revised Appendix.
> > We found that an unfine-tuned LLM can effectively correct "False" triplets for two reasons. First, we performed a new ablation study to show how the Revise action can substantially improve the agent’s performance, as shown in the Table below.
> >
> > ---
> > | Datasets | MMLU-Med | MedQA-US | PubMedQA* | BioASQ-Y/N | MedDDx-Basic | MedDDx-Intermediate | MedDDx-Expert|
> > |----------|----------|----------|-----------|------------|--------------|---------------------|--------------|
> > |KGARevion (w/o Revise)| 0.657 | 0.594 | 0.562 | 0.723 | 0.386 | 0.372 | 0.327|
> > |KGARevion | 0.703 | 0.628 | 0.590 | 0.744 | 0.473 | 0.451 | 0.411 |
> > |Improvement | 4.6% | 3.4% | 2.8% | 2.1% | 8.7% | 7.9% | 8.4% |
> > ---
> >
> >
> > Second, the LLM can revise a triplet when provided with a "False" triplet. The Revise action is solely responsible for modifying the False triplet to generate a corrected version. Once the revised triplet is created, it is passed to the Review action for validation to ensure its correctness.
> >
> > ## Q2 Incomplete Knowledge
> > Thank you for this important question that we consider in the manuscript. Specifically, triplets involving entities not included in the knowledge graph, identified as incomplete knowledge, are retained and incorporated into the Answer agent. These triplets are used alongside verified triplets to generate the final answer, ensuring that potentially valuable information is not overlooked.

---

> > > ### Comment · Reviewer_pKDu · 2024-11-26
> > >
> > > Thanks to the author. I think the author's additions to the Methods section and the Appendix explained my question well and helped improve the paper. I think the score I gave now reflects the quality of the paper well.

---

### Official Review · Reviewer_yzKB · 2024-10-30

**Soundness:** 4
**Presentation:** 3
**Contribution:** 3
**Rating:** 6
**Confidence:** 4

**Summary:**

The paper proposes a novel knowledge graph-based prompting technique for question answering in the medical domain. The approach consiststs of four steps: (1) Given a question in natural language, an LLM is used to generate subject-predicate-object triples from the question. (2) The triples are checked by means of a knowledge graph. Triples that are not clearly wrong are kept. (3) Then an LLM is asked to revise the triples. (4) Finally, the question is answered by means of an LLM and the revised triples.

**Strengths:**

- The paper tackles an important problem: question answering in the medical domain
- Original, novel question answering architecture: (1) generate triples, (2) review triples, (3) revise triples, (4) answer question
- Strong evaluation: multiple datasets, evaluation depending on question complexity, and ablation study
- Clarity: high-quality figures that help understanding

**Weaknesses:**

- Section 3.2 "fine-tuning stage" I find this part rather hard to understand. Figure 2b definitely helps, but it might be good to clarify this part even further by providing an example.
- Section 4.4: It suprises me that the results barely depend on the knowledge graph. I am wondering whether the knowledge graph is required at all then. What do the results look like for a knowledge graph that is not specialized to medicine (e.g., Freebase15k-237)? If the results for Freebase15k-237 are almost the same as for PrimeKG, then the KG would be irrelevant.
- Related work on fact-checking seems to be missing.

Ciampaglia, Giovanni Luca, Prashant Shiralkar, Luis M. Rocha, Johan Bollen, Filippo Menczer, and Alessandro Flammini. "Computational fact checking from knowledge networks." PloS one 10, no. 6 (2015): e0128193.

Shi, Baoxu, and Tim Weninger. "Discriminative predicate path mining for fact checking in knowledge graphs." Knowledge-based systems 104 (2016): 123-133.

**Questions:**

- Can you give an example for the "fine-tuning stage" in Section 3.2?
- Section 3.2 "inference stage": How often does each case occur (triple removed vs triple kept) for the given dataset?
- Can you run your question answering approach with a knowledge graph that is mostly unrelated to the questions (e.g., Freebase15k-237)? Do the results get significantly worse or do they stay approximately the same?
- In how far does each component of the approach handle and support multi-hop paths in knowledge graphs? It seems, the generate and revise steps do not support paths. The review step might support paths indirectly via KG embeddings.
- How does the review phase of the approach compare to fact-checking approaches?
- The approach seems rather general. Wouldn't it also work on other domains besides medicine?

---

> ### Author Response · Authors · 2024-11-20
> **Response to Weakness 1**
>
> ### Weakness 1 Writing in Fine-tuning Stage
>
> We apologize if these points were unclear in the paper's original presentation. We worked to clarify this section so it is easier to follow. We have refined this section as follows:
>
> “
> ###  *3.2 Review Action*
> *To enable LLMs to judge the correctness of generated triplets, beyond relying solely on semantic dependencies inferred by LLMs (Shinn et al., 2023), the Review action leverages relationships among various medical concepts in KGs. This is achieved by fine-tuning the LLM on a KG completion task and by explicitly integrating entity embeddings learned from KGs into the LLM. Then, the Review action is performed by the fine-tuned LLM to assess the correctness of triplets generated by the Generate action\.*
>
> ### *3.2.1 Fine-tuning Stage*
> ***Generating KG Embeddings and Triplet Descriptions:***   *We use TransE (Bordes et al., 2013), a well-known KG representation learning method, to learn embeddings for entities and relations in $G$. For a triplet $(h, r, t) \\in G$, the learned pre-trained KG embeddings are denoted as $\\textbf{e}\_h \\in \\mathbb{R}^{d}$, $\\textbf{e}\_r \\in \\mathbb{R}^{d}$, and $\\textbf{e}\_t \\in \\mathbb{R}^{d}$, where $d$ represents the embedding dimension. These embeddings are kept fixed.*
>
> *Additionally, we instruct the LLM to generate a description template for each relation $r \\in G$. We store these descriptions in a dictionary $D(r)$, where the key is the relation and the value is its description. The description of the dictionary can be found in Appendix Table 8\.*
>
> ***Aligning Embeddings:***  *Since LLM embeddings are based on token vocabularies (Radford et al., 2019), LLMs cannot directly use pre-trained KG embeddings. To use the pre-trained KG embeddings, we align them with the corresponding descriptions to generate new embeddings for each input triplet.*
>
> *Specifically, given the description $D(r)$ for the input triplet $(h, r, t)$, we denote the embedding of $D(r)$ obtained from the LLM as $\\mathbf{X} \\in \\mathbb{R}^{\\vert l \\vert \\times d\_L}$, where $\\vert l \\vert$ is the maximum number of tokens and $d\_L$ is the embedding dimension in the LLM.*
>
> *Next, we concatenate the embeddings of the head entity, relation, and tail entity, denoted as $\\mathbf{V} \= \[g(\\mathbf{e}\_h); g(\\mathbf{e}\_r); g(\\mathbf{e}\_t)\] \\in \\mathbb{R}^{3 \\times d\_L}$, where $g(\\cdot): \\mathbb{R}^d \\rightarrow \\mathbb{R}^{d\_L}$.We then apply an attention block (Vaswani, 2017), followed by a two-layer feedforward neural network (FFN, as shown in Fig. 2b) to obtain the aligned triplet embedding matrix $\\mathbf{Z} \\in \\mathbb{R}^{3 \\times d\_L}$ as follows:*\\begin{equation}
>      \\widehat{\\mathbf{V}} \= \\mathbf{V} \+ \\sigma(\\mathbf{V}\\mathbf{X}^{T})\\mathbf{X}
> \\end{equation}  \\begin{equation}
>     \\mathbf{Z} \= \\widehat{\\mathbf{V}} \+ ((\\varphi(\\widehat{\\mathbf{V}})\\mathbf{W}\_{1}))\\mathbf{W}\_{2}
> \\end{equation}  *where $\\sigma(\\cdot)$ is the Softmax function, $\\varphi(\\cdot)$ represents layer normalization, $\\mathbf{W}\_1 \\in \\mathbb{R}^{d\_L \\times d\_h}$ and $\\mathbf{W}\_2 \\in \\mathbb{R}^{d\_h \\times d\_L}$ are trainable parameters in the two-layer FFN, and $d\_h$ is the dimension of the hidden layer in the FFN.*
>
> ***Fine-tuning LLM:***  *After obtaining the aligned embedding $\\mathbf{Z}$, we add it to the beginning of the instruction and fine-tune the LLM using LoRA (Hu et al., 2022) with the next-token prediction loss (Radford, 2018). The instruction is: 'Given a triplet from a knowledge graph, where each triplet consists of a head entity, a relation, and a tail entity, please determine if the triplet is correct and respond with True or False.'*
>
> ### *3.2.2 Inference Stage*
> *The fine-tuned LLM is then used in the Review action to check the accuracy of each triplet in $T$, which was generated by the Generate action (3.1). Specifically, we first use UMLS codes (Bodenreider, 2004) to map the entities in the KG and obtain pre-trained KG embeddings for the head entity, relation, and tail entity, respectively. These embeddings, along with the relation description and the instruction, are fed into the fine-tuned LLM to determines whether each triplet is correct or not.*
>
> *However, not all entities in the generated triplet $(h, r, t) \\in T$ can be mapped to entities in KGs. To address this, the Review action applies a soft constraint rule to distinguish whether the generated triplet is factually wrong or the result of incomplete knowledge in KGs, as follows:*
> - *Factually Wrong: if we can map $h$ and $t$ to entities in KGs and the output of fine-tuned LLM is False, then the triplet $(h, r, t)$ is factually wrong and is removed from $T$.*
> - *Incomplete Knowledge: if we cannot map either $h$ or $t$ to entities in KGs, then the triplet $(h, r, t)$ is considered incomplete knowledge and is kept.*
> ”

---

> ### Author Response · Authors · 2024-11-20
> **Response to Weakness 2**
>
> ### Weakness 2 Utility of Knowledge Graph
>
> Thank you for bringing up this point, which we should clarify. We consider PrimeKG and OGB-biokg in our experiments because these two knowledge graphs are widely adopted in the field and have a broad representation of biomedicine. Results in the paper suggest that KGARevion is flexible in that the agent can be used with various KGs and that the agent is not apriori tied to any specific KG. Also, model performance remains strong with the agent using either PrimeKG or OGB-bioKG (see Section 4.4 on the versatility of KGARevion).
>
> However, your comment points to another important and exciting question, which is the coverage of the knowledge domain (i.e., medical vs non-medical KGs). To address your question, we explored running experiments using non-medical Freebase15k-237 KG. However, upon closer inspection, we found that the Freebase KG does not cover at all medical domains, let alone specialized medical knowledge of molecules or medical concepts. Based on KGARevion’s design, if we cannot map entities in the generated triplets to entities in the knowledge graph, those triplets are still retained as "Incomplete Knowledge" and used to generate the final answers. As a result, using an irrelevant KG would lead to outcomes similar to removing the Review action entirely. To test this claim, we performed a new experiment where we excluded the Review action from KGARevion. Results of this new experiment are shown in the Table below:
>
> | Datasets | MMLU-Med | MedQA-US | PubMedQA\* | BioASQ-Y/N | MedDDX-Basic | MedDDx-Intermediate | MedDDx-Expert |
> | :---- | :---- | :---- | :---- | :---- | :---- | :---- | :---- |
> | KGARevion(w/o Review) | 0.695 | 0.546 | 0.560 | 0.736 | 0.298 | 0.299 | 0.327 |
> | KGARevion | 0.734 | 0.620 | 0.638 | 0.763 | 0.483 | 0.457 | 0.447 |
> | Improvement | 3.9% | 7.4% | 7.8% | 2.7% | 18.5% | 15.8% | 12% |
>
> Based on these results, we conclude that grounding triplets in biomedical KGs is valuable. However, even if KGARevion were used with an irrelevant or out-of-domain KG, it would not experience catastrophic degradation. In such cases, the agent would simply skip the Review action. In the extreme scenario where the KG provides no useful information, the system would rely entirely on the LLM as a fallback.

---

> ### Author Response · Authors · 2024-11-20
> **Response to Weakness 3**
>
> ### Weakness 3 Related Work
>
> Thank you for your valuable suggestion. We have incorporated the following related work into our latest version.
>
> ***“Fact-checking.** The vast amount of generated text has created an increasing need for fact-checking (Graves & Glaisyer, 2012). Early methods addressed this challenge by relying on manually defined rules to verify generated statements (Hassan et al., 2014; Wu et al., 2012; Jiang et al., 2011). KGs have also been leveraged for fact-checking through graph connectivity analysis. These methods validate statements by identifying paths between the entities mentioned (Tunstall-Pedoe, 2010; Shi & Weninger, 2016; Ciampaglia et al., 2015). Furthermore, some approaches use KG representation learning algorithms for KG completion tasks to assess the accuracy of statements (Li et al., 2011). With the advent of LLMs, novel fact-checking techniques have emerged. For example, retrieval-augmented generation (RAG)-based models integrate external evidence retrieval to ensure the consistency and reliability of LLM-generated outputs by verifying and supporting their content (Asai et al., 2024).*
>
> *References:*
>
> - *Graves L, Glaisyer T. The fact-checking universe in Spring 2012\[J\]. New America, 2012\.*
> - *Naeemul Hassan, Afroza Sultana, You Wu, Gensheng Zhang, Chengkai Li, Jun Yang, and Cong Yu. Data in, fact out: Automated monitoring of facts by factwatcher. Proceedings of the VLDB Endowment, 7(13):1557–1560, 2014\.*
> - *You Wu, Pankaj K Agarwal, Chengkai Li, Jun Yang, and Cong Yu. On” one of the few” objects. In Proceedings of the 18th ACM SIGKDD international conference on Knowledge discovery and data mining, pp. 1487–1495, 2012\.*
> - *Xiao Jiang, Chengkai Li, Ping Luo, Min Wang, and Yong Yu. Prominent streak discovery in sequence data. In Proceedings of the 17th ACM SIGKDD international conference on Knowledge discovery and data mining, pp. 1280–1288, 2011\.*
> - *William Tunstall-Pedoe. True knowledge: Open-domain question answering using structured knowledge and inference. Ai Magazine, 31(3):80–92, 2010\.*
> - *Baoxu Shi and Tim Weninger. Discriminative predicate path mining for fact checking in knowledge graphs. Knowledge-based systems, 104:123–133, 2016\.*
> - *Giovanni Luca Ciampaglia, Prashant Shiralkar, Luis M Rocha, Johan Bollen, Filippo Menczer, and Alessandro Flammini. Computational fact checking from knowledge networks. PloS one, 10(6):e0128193, 2015\.*
> - *Xian Li, Weiyi Meng, and Clement Yu. T-verifier: Verifying truthfulness of fact statements. In 2011IEEE 27th International conference on data engineering, pp. 63–74. IEEE, 2011\.*
> - *Akari Asai, Zeqiu Wu, Yizhong Wang, Avirup Sil, and Hannaneh Hajishirzi. Self-RAG: Learning to retrieve, generate, and critique through self-reflection. In The Twelfth International Conference on Learning Representations, 2024\. URL https://openreview.net/forum?id=hSyW5go0v8.”*

---

> ### Author Response · Authors · 2024-11-20
> **Response to Questions**
>
> ### Q1 Example for Fine-tuning
>
> Taking the input “*(HSPA8, interacts, DHDDS)*” as an example:
>
> - Get the embedding of “*HSPA8”*, “*interacts”*, “*DHDDS”*, respectively. These embeddings are learned from the KG;
> - Get the description for the triplet. The description is “*HSPA8 interactions with DHDDS, indicating a general interaction that may involve binding, modulation, or other forms of molecular communication*”;
> - Get aligned embeddings. We align the embeddings generated by KGs with the embeddings of description obtained from LLMs to produce aligned embeddings for (“*HSPA8*”, “*interacts*”, “*DHDDS*”);
> - Finally, we send the aligned embeddings for (“*HSPA8”*, “*interacts”*, “*DHDDS”*) along with an instruction into LLM and fine-tune it with the next token prediction objective.
>
> ### Q2 Removed vs. Kept Triples
>
> Thank you for your suggestion. To address your comment, we have conducted new experiments to show the average number of removed vs. kept triplets. The results are shown in the following Table.
>
> | Datasets | MMLU-Med | MedQA-US | PubMedQA\* | BioASQ-Y/N | MedDDX-Basic | MedDDx-Intermediate | MedDDx-Expert |
> | :---- | :---- | :---- | :---- | :---- | :---- | :---- | :---- |
> | Generated Triplets | 14.1 | 19.2 | 2.9 | 2.8 | 12.4 | 12.1 | 12.1 |
> | Kept Triplets | 9.1 | 11.8 | 2.5 | 1.7 | 6.2 | 5.9 | 5.2 |
> | Removed Triplets | 5.0 | 7.4 | 0.4 | 1.1 | 6.2 | 6.2 | 6.9 |
>
> The numbers in this table show how often each case (removed vs. kept triplets) occurs on average across all queries in each dataset.
>
> ### Q3  Experiments with Irrelevant Knowledge Graph
>
> Thank you for your great point and suggestion. Please refer to our response in ***Weakness 2 Irrelevant Knowledge Graph***, where we report on new experiments and discussion addressing this question.
>
> ### Q4 Reasoning Paths
>
> Thank you for your valuable comment. KGARevion supports paths indirectly through KG embeddings. Additionally, it supports multi-hop reasoning in the Answer action by considering all verified triplets (which can form a multi-hop reasoning path) when generating the final answer.
>
> ### Q5 Review vs. Fact-checking
>
> Thank you for your valuable suggestion and for sharing fact-checking approaches with us. We agree with the concept behind existing fact-checking models, which ground generated text on structured, accurate knowledge graphs. We apply this idea when designing the review action in KGARevion.
>
> Our focus is specifically on improving the LLMs’ ability to use medical reasoning by enabling them to verify the correctness of generated triplets, not only using the general knowledge within LLMs but also the structural knowledge from knowledge graphs.
>
> ### Q6 Applications
>
> Thank you for your kind words. One key aspect of KGARevion is its flexibility. In this work, we have shown that KGARevion performs well in expert-level, domain-specific medical reasoning. Based on our results, KGARevion could also be considered for other fields where domain-specialized and expert-level reasoning is needed.
>
> ---
> *Thank you for your helpful feedback. If you feel our responses are insufficient to motivate increasing your score, we would love to hear from you further about how we can better address your questions. Thank you again!*

---

> ### Author Response · Authors · 2024-11-28
>
> Thank you for your helpful feedback. We worked hard to improve our paper, and we sincerely hope the reviewers find our responses informative and helpful. If you feel the responses have not addressed your concerns to motivate increasing your score, we would love to hear what points of concern remain and how we can improve our work. Thank you again!

---

> > ### Comment · Reviewer_yzKB · 2024-11-29
> >
> > Thank you for answering my questions. After reading all the reviews and the authors' responses, I kept my score.

---

> ### Author Response · Authors · 2024-11-29
> **Author Response**
>
> Thank you for your response! If you have any more questions, I’d be happy to discuss them. If not, could you please consider improving your score to reflect the improvements and clarifications in the rebuttal?

---

### Official Review · Reviewer_jrR1 · 2024-10-30

**Soundness:** 2
**Presentation:** 3
**Contribution:** 3
**Rating:** 6
**Confidence:** 4

**Summary:**

This article introduces KGARevion, an LLM which performs medical Q&A by verifying generated information (in the form of triplets) against the information in a knowledge graph (provided as knowledge graph embeddings). By doing so, it achieves competitive performance on several medical Q&A benchmarks. Additionally, the authors provide a new biomedical Q&A benchmark, which they call MedDDx. MedDDx leverages information from another Q&A dataset, but it is supplemented with several incorrect answers which are semantically similar to the correct answers.  KGARevion can be used with a variety of base LLMs and KGs, making it not only competitive but also versatile.

**Strengths:**

- **Figures:** the figures are nice for understanding the paper; in particular, figure 2 is great.

- **Creative Benchmark:** the idea to create a benchmark with incorrect answers that are semantically similar to correct answers is very creative. In addition, separating questions into levels of difficulty is also quite clever.

- **Reproducibility:** I commend the authors for providing code. Although I did not test it, it looks well documented and well written.

- **Versatility:** the KGARevion approach seems versatile, able to be used with other base LLMs, KGs, and other domains in general.

**Weaknesses:**

Following the authors' rebuttal, I feel that my concerns below were sufficiently addressed. The authors have added an additional benchmark dataset from a sufficiently different data lineage, and they have also clarified several points of confusion. I have updated my score accordingly.

-------------------------------------------------------------------------------

I have a few concerns about the scientific robustness and the presentation of this work:

1. Firstly, my greatest concern regards the overlap between information in the knowledge graph(s) (KGs) used and the information used to create the benchmarks. Specifically, the authors designed their new QA benchmark, MedDDx, based upon the STaRK-Prime benchmark. **However, according to the corresponding STaRK-Prime paper (https://arxiv.org/abs/2404.13207), the biomedical QA from STaRK-Prime was built upon the PrimeKG, the exact KG which the authors used to fine-tune their approach, KGARevion.** Ultimately, this is not scientifically robust as the authors are providing their LLM with the same, structured data on which their benchmark was built.

    I understand that the authors have also used other "gold-standard" benchmarks for comparison. However, there is also likely data leakage in some of these cases as well. For example, the PrimeKG is built from structured data derived from other sources, including the Disease Ontology (DO), the Gene Ontology (GO), UMLS, and DrugBank (Fig. 2, https://www.nature.com/articles/s41597-023-01960-3), all of which are also included in the creation of the BioASQ benchmark (https://pmc.ncbi.nlm.nih.gov/articles/PMC10042099/). Unfortunately, the other KG used within Section 4.4, ogb-bioKG, does not have any provenance information (https://github.com/snap-stanford/ogb/issues/111) but it was built by the same person (M. Zitnik) as PrimeKG, so it likely comes from similar sources. Ultimately, I would still avoid using obg-bioKG if the provenance can not be verified.

    **I recommend that the authors research the data lineage of each of the benchmarks and KGs used. Perhaps they can present this as supplementary material. Then, I recommend that the authors only present the results in which there is no clear overlap between the structured data included within the KGs for fine-tuning and the benchmark datasets. The authors should omit all results where data leakage between the KGs and benchmarks is certain or likely.**



2. Secondly, I found this paper generally difficult to follow. Specifically, I believe the study could be more clearly motivated within the introduction, and Section 3 ("Approach") could be more clearly explained. Below, I list some specific points for improvement:

- Paragraph 1 and the first half of paragraph 2 of the introduction seem loosely connected to the rest of the motivation; I suggest cutting these paragraphs and skipping more directly to the points in paragraph 3.

- Figure 1 is problematic for several reasons. First, it is presented in the middle of the introduction, which is an inappropriate place to present results. Secondly, it is presented with no experimental context. Finally, it is not clear whether the figure even supports the claims being made in the introduction. I suggest moving or removing this figure.

- RAG systems are presented as less desirable alternatives due to a reliance upon the quality of the documents provided, but the authors acknowledge that KGs may also have incorrect or incomplete information. Therefore, the motivation for the work seems unclear.

- Section 3.2 on the Review section is extremely difficult to read. I believe this could be made clearer through more structure within the subsection (sub-headings or numbered steps). Perhaps the textual structure could correspond to flagposts or landmarks within Figure 2, so that the reader can refer back and forth between the text and Figure 2 easily.

- Although fine-tuning with a KG is an integral part of the methodology, PrimeKG is not mentioned anywhere within Section 3. Until I read the Supplementary materials, I had understood that the KG being used for fine-tuning was the set of triplets derived from the "Generate" action. The authors should make it clearer that the KG comes from another, external source.

**Questions:**

1. page 3, Section 3: The authors give an example of a possible question and a set of possible answers. However, wouldn't this question warrant a yes/no response, rather than the set of answers provided?

2. page 5: On which prediction task was TransE trained?

3. page 5: Why have the authors chosen TransE for the KG embeddings?

4. page 5: Where does the description dictionary D(r) come from? I do not see this in the appendix, and there is no appendix section 8.

5. page 6, section 3.3: How, specifically, does the Revise action "adjust the triplets in F to include more knowledge"?

6. page 8, Fig. 3: What do the blue v. orange plots mean? The only y-axis label I see is accuracy, and I see no key for blue v. orange.

---

> ### Author Response · Authors · 2024-11-19
> **Global Response**
>
> Thank you for your valuable and thoughtful feedback. We sincerely appreciate your kind reviews on the figures and benchmarks and how reproducible and versatile our approach is. We also appreciate your constructive suggestions regarding our writing, datasets, and methodological details. To address your concerns, we have carefully responded to each point one-by-one. **Most importantly, we have included a number of new experiments and added new datasets and benchmarks, where we can guarantee that there is i) no information leakage from the LLM into the newly added benchmark, as well as that there is ii) no information leakage from the KG into the newly added benchmark.**
>
> **We respond to all your comments below. If you feel we have not sufficiently addressed your concerns to motivate increasing your score, we would love to hear from you further on what points of concern remain and how we can improve the work in your eyes. Thank you again\!**

---

> ### Author Response · Authors · 2024-11-19
> **Responses to Weakness 1**
>
> ### Weakness 1 Data Leakage
> Thank you for pointing out this area that we should clarify. Our response to this critical question has two parts. First, we present new experiments and benchmarks. Second, we offer a clarification on benchmarks and results included in the initial submission of this paper.
>
> **Part 1**
> To address your concerns with absolute certainty that there is no leakage, we have included another newly published QA dataset, AfriMed-QA (https://huggingface.co/datasets/intronhealth/afrimedqa\_v2), as a benchmark.
>
> - AfriMed-QA creates a novel multispecialty open-source dataset of 15,000 pan-African clinically diverse QA to rigorously evaluate LLMs in African healthcare.
> - The dataset is sourced from over 500 clinical and non-clinical contributors across 16 countries and covers 32 clinical specialties.
> - **First, because the dataset is sourced from healthcare systems whose data are not online, it means that none of the multiple-choice and open-ended questions and answers can be in the knowledgebase of the LLM. Because of that, this benchmark has not been leaked into KGARevion’s LLM. Furthermore, the knowledge cut-off date for KGARevion’s LLM was before AfriMed-QA was released.**
> - **Second, AfriMed-QA is sourced from clinical, private healthcare systems whereas PrimeKG, STaRK-Prime, and OGB-bioKG are sourced from non-clinical, public, biological data repositories. Furthermore, PrimeKG, STaRK-Prime, and OGB-bioKG were all released several years prior to when the AfriMed-QA benchmark was released. As such, AfriMed-QA has certainly not been leaked into KGARevion’s KG**.
>
> **In summary, we can now guarantee that there is no data leakage from the newly added AfriMed-QA benchmark into KGARevion because:**
>
> - The queries in AfriMed-QA are entirely **independent** of the underlying KG, such as PrimeKG.
> - The answers to AfriMed-QA queries are **verifiably absent from the KG or the LLM's knowledge base**.
> - AfriMed-QA was released on September 12, 2024, after the publication of LLaMA3.1-8B on July 18, 2024\.
>
> Please let us know if that is the case, or if you have any other concerns that remain on this point. We fully agree that the issue of data leakage is incredibly important and recognize that many recently published papers suffer from the issue of potential leakage. We are grateful for a chance to clarify this point and include additional results that are scientifically robust and for which we can verify that there is no leakage from training into test sets.
>
> From the 15,000 queries in the AfriMed-QA, we first selected 3,910 multiple-choice queries labeled ‘test’ and ‘Expert’ in the dataset. To save time, we tested GPT-4-Turbo on all 3,910 queries and identified 910 queries that could not be answered correctly. We then tested KGARevion and the baseline models on these 910 queries. The results are shown below:
>
> | Methods | KG-Rank  | Self-RAG (LLaMA2-13B) | LLaMA3-8B | LLaMA3.1-8B | KGARevion |
> | :---- | :---- | :---- | :---- | :---- | :---- |
> | Accuracy | 0.136 | 0.205 | 0.195 | 0.209 | 0.256 |
> | Improvement | 12% | 5.1% | 6.1% | 4.7% | \- |
>
> These results further indicate the effectiveness of KGARevion, providing additional evidence that KGARevion is a well-performing KG-based agent and that KGARevion is scientifically robust. We will make sure to clarify these points in the revised manuscript and integrate these new experiments in the Results section of the paper.
>
> **Part 2**
> **We think there may be a misunderstanding here, for which we apologize. While we will clarify the text on this point, we would like to highlight that in the case of benchmarks included in the initial submission, there is no leakage** **in evaluating the correctness of generated triplets in KGARevion** for the following reasons:
>
> - Both ‘Generate’ and ‘Revise’ actions are performed by a standalone LLM, e.g., LLaMA-3.1-8B, by instructing the LLM in generating triplets related to the input query or revising triplets, respectively.
> - The fine-tuning of the LLM on the knowledge graph completion task is designed to enable the LLM to leverage structural embeddings learned from the KG, e.g., PrimeKG, to assess the correctness of triplets. This fine-tuned LLM carries out the Review action. It should be noted that the triplets used in fine-tuning the LLM are from PrimeKG.
> - Crucially, the Review action is specifically intended to evaluate the correctness of triplets **generated by the ‘Generate’ action**, **NOT those from PrimeKG or OGB-biokg**.
>
> We apologize if any of these points were unclear in the paper's original presentation, and will work hard to clarify so this message is better conveyed.
>
> We hope the new experiments in Part 1 and additional explanations in Part 2 provide sufficient evidence that KGARevion and evaluations are scientifically robust. Thank you again for raising this critical point. We will work hard to clarify the text and include new experiments in the paper.

---

> ### Author Response · Authors · 2024-11-19
> **Responses to Weakness 2.1-2.3**
>
> ### Weakness 2.1 Redundancy of Paragraphs 1&2 in Introduction
>
> We are sorry that the paper wasn’t clear, and we will work to clarify it, specifically focusing on flow, removing unnecessary abbreviations, and removing any redundant parts. We have removed it in the latest version and started directly with Paragraph 3\.
>
>
> ### Weakness 2.2 Figure 1
>
> Thank you for your suggestion. We have removed Figure 1a to the section of results.
>
>
> ### Weakness 2.3 RAG vs. KG
>
> Thank you for your comment. To clarify, our claim refers only to RAG systems potentially suffering from incorrect or incomplete information, not KGs. We aim to use the structured and accurate knowledge in KGs to improve LLM performance in medical QA and enhance its reasoning ability. In KGARevion, the KG is not used for knowledge retrieval like that in RAG systems but as a verifier to review the triplets generated by a standalone LLM.

---

> ### Author Response · Authors · 2024-11-19
> **Response to Weakness 2.4**
>
> ### Weakness 2.4 Writing on Review
> Thank you for pointing out this area that we should clarify. We have refined the architecture of the Review action, as shown below:
>
> “
> ###  *3.2 Review Action*
> *To enable LLMs to judge the correctness of generated triplets, beyond relying solely on semantic dependencies inferred by LLMs (Shinn et al., 2023), the Review action leverages relationships among various medical concepts in KGs. This is achieved by fine-tuning the LLM on a KG completion task and by explicitly integrating entity embeddings learned from KGs into the LLM. Then, the Review action is performed by the fine-tuned LLM to assess the correctness of triplets generated by the Generate action\.*
>
> ### *3.2.1 Fine-tuning Stage*
> ***Generating KG Embeddings and Triplet Descriptions:***   *We use TransE (Bordes et al., 2013), a well-known KG representation learning method, to learn embeddings for entities and relations in $G$. For a triplet $(h, r, t) \\in G$, the learned pre-trained KG embeddings are denoted as $\\textbf{e}\_h \\in \\mathbb{R}^{d}$, $\\textbf{e}\_r \\in \\mathbb{R}^{d}$, and $\\textbf{e}\_t \\in \\mathbb{R}^{d}$, where $d$ represents the embedding dimension. These embeddings are kept fixed.*
>
> *Additionally, we instruct the LLM to generate a description template for each relation $r \\in G$. We store these descriptions in a dictionary $D(r)$, where the key is the relation and the value is its description. The description of the dictionary can be found in Appendix Table 8\.*
>
> ***Aligning Embeddings:***  *Since LLM embeddings are based on token vocabularies (Radford et al., 2019), LLMs cannot directly use pre-trained KG embeddings. To use the pre-trained KG embeddings, we align them with the corresponding descriptions to generate new embeddings for each input triplet.*
>
> *Specifically, given the description $D(r)$ for the input triplet $(h, r, t)$, we denote the embedding of $D(r)$ obtained from the LLM as $\\mathbf{X} \\in \\mathbb{R}^{\\vert l \\vert \\times d\_L}$, where $\\vert l \\vert$ is the maximum number of tokens and $d\_L$ is the embedding dimension in the LLM.*
>
> *Next, we concatenate the embeddings of the head entity, relation, and tail entity, denoted as $\\mathbf{V} \= \[g(\\mathbf{e}\_h); g(\\mathbf{e}\_r); g(\\mathbf{e}\_t)\] \\in \\mathbb{R}^{3 \\times d\_L}$, where $g(\\cdot): \\mathbb{R}^d \\rightarrow \\mathbb{R}^{d\_L}$.We then apply an attention block (Vaswani, 2017), followed by a two-layer feedforward neural network (FFN, as shown in Fig. 2b) to obtain the aligned triplet embedding matrix $\\mathbf{Z} \\in \\mathbb{R}^{3 \\times d\_L}$ as follows:*\\begin{equation}
>      \\widehat{\\mathbf{V}} \= \\mathbf{V} \+ \\sigma(\\mathbf{V}\\mathbf{X}^{T})\\mathbf{X}
> \\end{equation}  \\begin{equation}
>     \\mathbf{Z} \= \\widehat{\\mathbf{V}} \+ ((\\varphi(\\widehat{\\mathbf{V}})\\mathbf{W}\_{1}))\\mathbf{W}\_{2}
> \\end{equation}  *where $\\sigma(\\cdot)$ is the Softmax function, $\\varphi(\\cdot)$ represents layer normalization, $\\mathbf{W}\_1 \\in \\mathbb{R}^{d\_L \\times d\_h}$ and $\\mathbf{W}\_2 \\in \\mathbb{R}^{d\_h \\times d\_L}$ are trainable parameters in the two-layer FFN, and $d\_h$ is the dimension of the hidden layer in the FFN.*
>
> ***Fine-tuning LLM:***  *After obtaining the aligned embedding $\\mathbf{Z}$, we add it to the beginning of the instruction and fine-tune the LLM using LoRA (Hu et al., 2022) with the next-token prediction loss (Radford, 2018). The instruction is: 'Given a triplet from a knowledge graph, where each triplet consists of a head entity, a relation, and a tail entity, please determine if the triplet is correct and respond with True or False.'*
>
> ### *3.2.2 Inference Stage*
> *The fine-tuned LLM is then used in the Review action to check the accuracy of each triplet in $T$, which was generated by the Generate action (3.1). Specifically, we first use UMLS codes (Bodenreider, 2004) to map the entities in the KG and obtain pre-trained KG embeddings for the head entity, relation, and tail entity, respectively. These embeddings, along with the relation description and the instruction, are fed into the fine-tuned LLM to determines whether each triplet is correct or not.*
>
> *However, not all entities in the generated triplet $(h, r, t) \\in T$ can be mapped to entities in KGs. To address this, the Review action applies a soft constraint rule to distinguish whether the generated triplet is factually wrong or the result of incomplete knowledge in KGs, as follows:*
> - *Factually Wrong: if we can map $h$ and $t$ to entities in KGs and the output of fine-tuned LLM is False, then the triplet $(h, r, t)$ is factually wrong and is removed from $T$.*
> - *Incomplete Knowledge: if we cannot map either $h$ or $t$ to entities in KGs, then the triplet $(h, r, t)$ is considered incomplete knowledge and is kept.*
> ”

---

> ### Author Response · Authors · 2024-11-19
> **Responses to Weakness 2.5**
>
> ### Weakness 2.5 Details in the Fine-tuning Stage in Review
>
> We apologize for any ambiguity. We have added the following to the last paragraph:
>
> '*Our KGARevion is compatible with any medical-related KGs (e.g., PrimeKG, OGB-biokg) and LLMs (e.g., LLaMA3-8B, LLaMA3.1-8B).*'

---

> ### Author Response · Authors · 2024-11-19
> **Responses to Questions**
>
> ### Q1 Example in Section3
> Thank you for your careful review, and we apologize for that.
>
> We have revised it to “*Which gene interacts with the Heat Shock Protein 70 family that acts as a molecular chaperone, and is implicated in Retinitis Pigmentosa 59 due to DHDDS mutation?*”
>
> ### Q2 Prediction Task for TransE
>
> Thank you for your comment. TransE is pre-trained on PrimeKG on the node representation learning task in a self-supervised way.
>
> ### Q3 Reason for TransE
>
> TransE is a well-established and popular KG representation learning method and has advantages in capturing the relationship between entities in KG (Liu et al., 2015). For this reason, we chose TransE for our work.
>
> References:
> Lin Y, Liu Z, Sun M, et al. Learning entity and relation embeddings for knowledge graph completion\[C\]//Proceedings of the AAAI conference on artificial intelligence. 2015, 29(1)
>
> ### Q4 Sources for *D(r)*
>
> The description is generated by the LLM based on the following instruction:
> '*Given a relationship between molecules, generate a description for the input relationship.*'
>
> The description dictionary can be found in **Appendix Table 8**, not Section 8\. Apologies for the typo.
>
> ### Q5 Expression on the Revise Action
>
> We apologize for the confusion. The goal of the Revise action is to expand the number of triplets related to the input query, improving the coverage of its concepts.
>
> The Revise action instructs a standalone LLM to revise an incorrect (‘False’) triplet by replacing either the head or tail entity. For example:
>
> *'The following triplet is incorrect: (HSPA1A, interactions, DHDDS). Please revise it to a correct triplet relevant to the input query.'*
>
> The LLM might then generate a new triplet, such as (HSPA1B, interactions, DHDDS), by replacing the head entity.
>
> In this way, the Revise action increases the available triplets and ensures better coverage of the medical concepts in the input query.
>
> ### Q6 Color in Figure 3
>
> The difference between blue and orange lies in the backbone LLM: blue represents LLaMA3-8B, while orange represents LLaMA3.1-8B.
>
> ---
> *Thank you again for your thoughtful commentary. If you feel our responses are insufficient to motivate increasing your score, we would love to hear from you further about how we can better address your concerns. Thank you again\!*

---

> > ### Comment · Reviewer_jrR1 · 2024-11-24
> > **Response to Author Rebuttal**
> >
> > Thanks to the authors for the detailed rebuttal. I appreciate the effort put in, and I will change my review accordingly.

---

> ### Author Response · Authors · 2024-11-24
>
> Dear Reviewer jrR1,
>
> We sincerely appreciate your insightful question and valuable suggestions during the author-reviewer phase. Those are extremely useful, and I believe this submission experience is the best one we've had so far. Please never hesitate to leave us a comment if you have further ideas during our interaction with other reviewers. Thanks again!
>
> Authors

---

### Official Review · Reviewer_XbfB · 2024-11-05

**Soundness:** 4
**Presentation:** 3
**Contribution:** 3
**Rating:** 6
**Confidence:** 4

**Summary:**

This paper studies the integration of factual knowledge with free text information into LLMs. It introduces KGAREVION, a medical Question Answering (QA) agent, which additionally fine-tunes a LLM on a knowledge base completion task to improve its performance on evaluating medical facts. For an input question the agent can perform three actions:
 1. **Generate** relevant medical fact candidates
 1. **Review** the generated fact candidates to check factual incorrect facts
 1. **Revise** to add more facts and change the factual incorrect candidate from Review
The outputs of the last action are used as part of a prompt to generate the final answer. The agent is compared on multiple choice and open ended questions against several baselines. It is able to reach state of the art performance.

**Strengths:**

**Medical Application**

The paper study the application of an LLM-based agent to a complex and difficult domain, viz. medicine. I appreciate that the authors took up this challenge to make a relevant contribution.

**Evaluation**

The presented approach is evaluated thoroughly. It covers a comparison against a large number of competitors on open-ended and multi-choice reasoning. It shows that the presented approach is able to reach state of the art performance and moves the bar.  Apart from this performance analysis the paper also offers an ablation study and sensitivity analyses.

**Weaknesses:**

Complex approach:

The presented approach is complex and requires a lot of steps and technical components. It depends on a separate entity recognizer to map entities to UMLS, two LLMs performing different tasks (generation and revision, review), and a knowledge graph for fine-tuning. This reduces reusability, since the entire system would need to be replicated. Reducing all this into one LLM and dataset would make the contribution for accessible for others.

**Questions:**

Did you consider using the facts and relations from UMLS for the KB completion task?

---

> ### Author Response · Authors · 2024-11-24
> **Response to Weakness and Question**
>
> ### Weakness 1 Complex Approach
>
> Thank you for your suggestions. KGARevion uses four main actions to support medical reasoning: **Generate**, **Review**, **Revise**, and **Answer**.
>
> 1. **Generate**: KGARevion uses the general knowledge in LLM to generate relevant triplets (subject-predicate-object) that could potentially answer the input query.
> 2. **Review**: KGARevion then grounds these generated triplets by cross-referencing them with a KG to verify their correctness.
> 3. **Revise**: Any triplets identified as "False" during the Review action are corrected by the LLM. The revised triplets are then sent back for further review, where they are re-evaluated until they are deemed correct or the maximum number of review rounds is reached.
> 4. **Answer**: KGARevion uses the grounded triplets to infer and provide the final answer to the input query.
>
> By combining the power of LLMs with a knowledge graph, KGARevion ensures that the generated answers are both contextually relevant and factually grounded.
>
> ### Entity recognizer
>
> We use UMLS to map entities in KG to improve KGARevion's flexibility. Different KGs may use different molecular vocabularies, so adopting a unified recognizer allows KGARevion to be applied across various KGs.
>
> ### The number of LLMs used in KGARevion
>
> KGARevion uses the same LLM as the backbone for all actions. The **Generate**, **Revise**, and **Answer** actions are handled by the base LLM (e.g., LLaMA3.1-8B), while the **Review** action uses a fine-tuned version of the same model.
>
> In implementation, the model is loaded just once. Only when the fine-tuned LLM is needed are the fine-tuned weights loaded. Additionally, the fine-tuned model can be used as a standalone module by other projects and contributors.
>
> ### Q1 New Knowledge Graph
>
> Thank you for your suggestion. Since KGARevion works with any medical-related KGs, we can explore using facts and relationships from UMLS for the KG completion task. To answer your question, we performed a new experiment using the following setup:
>
> 1. Download the latest version of UMLS from [this link](https://www.nlm.nih.gov/research/umls/licensedcontent/umlsknowledgesources.html). Note that UMLS contains key terminology, classification, and coding standards, as well as many vocabularies, including CPT, ICD-10-CM, LOINC, MeSH, RxNorm, and SNOMED CT.
> 2. Fine-tune a large language model (e.g., LLaMA3.1-8B) using the UMLS data for KG completion task. The LLM takes the KG embeddings from UMLS for each triplet and the instruction as input to determine whether each tripe is True or False. The LLM is fine-tuned by LoRA with next token prediction loss.
> 3. Integrate the fine-tuned LLM with KGARevion and test KGARevion (UMLS) on all datasets.
>
> The results we obtained are as follows:
>
> | Datasets | MMLU-Med | MedQA-US | PubMedQA\* | BioASQ-Y/N | MedDDX-Basic | MedDDx-Intermediate | MedDDx-Expert |
> | :---- | :---- | :---- | :---- | :---- | :---- | :---- | :---- |
> | LLaMA3.1-8B | 0.677 | 0.563 | 0.596 | 0.687 | 0.434 | 0.368 | 0.306 |
> | KGARevion (UMLS) | 0.691 | 0.603 | 0.600 | 0.756 | 0.473 | 0.416 | 0.396 |
> | Improvement (%) | 1.4% | 4.0% | 0.4% | 6.9% | 3.9% | 4.8% | 9.0% |
>
> ---
> *We are very grateful for your constructive feedback about KGARevion. We appreciate that you recognized the core contributions of our work, as well as the novelty of multiple components of this agent. Please reach out if you have any additional questions or need clarification about this work. Thank you\!*

---

> ### Author Response · Authors · 2024-11-28
>
> Thank you for your helpful feedback. We worked hard to improve our paper, and we sincerely hope the reviewers find our responses informative and helpful. If you feel the responses have not addressed your concerns to motivate increasing your score, we would love to hear what points of concern remain and how we can improve our work. Thank you again!

---

### Public Comment · ~Mingyu_Huang1 · 2024-11-14
**A Public Comment**

I have taken some time reading this manuscript as I am myself interested in AI4Healthcare. While I found certain aspects of this work to be valuable, I however had several concerns as listed below that I want to discuss with the authors.

**Evaluation (data leakage).** After reading this manuscript I have the same concern as **reviewer jrR1**. To add to his/her comments, using a fine-tuned LLM to enhance reasoning, while intuitive, would pose significant concern to the fairness of the evaluation results on general medical benchmarks due to potential data leakage. While I understand that this is too big a question to be thoroughly addressed in a conference paper, the authors could have either $i)$ attempted to quantitatively measure the extent of the data leakage [1], $ii)$ more carefully designed the experiments to avoid this, or $iii)$ at least thoroughly discussed this as an important thread to validity of the results (though given the direct impact of these on the validity of the evaluation, this may not be sufficiently convincing) [2]. Yet it seems that these are not presented in the current manuscript. This makes it hard for me to interpret the results: whether the reported improvement by KGARevion is actually because of the its technical advancement, or has been simply benefited from the data leakage?

**Reliability 1 (KG quality).** The authors criticized existing RAG-based approaches for being susceptible to the quality of retrieved information. However, KGARevion also heavily relies on an external KG to fine-tune the LLM, whose quality would affect the performance as well as reliability of the approach. This makes the point that the authors intend to argue seem questionable. Notably, there has been recent works showing that existing KG can be easily "poisoned" by factually incorrect information presented in the source data used for constructing it [3]. In light of this and the giant scale of existing KGs, it is highly possible that the KG used in KGARevion also contain various unreliable information. It remains unclear how prevalent are such errors and how would they effect the performance of KGARevion. I am interested in these questions as they are important for a system targeted at the health domain.

**Reliability 2 (LLM for entity extraction).** Another concern I have regarding reliability is the generation step (Section 3.1). Here the authors used an LLM for extracting medical triplets from the questions. Considering the well-known hallucination issues of the LLMs, it is very likely that the LLM used here may "invent" some entities that do not actually exist in the original question. The current manuscript does not touch on this. Also, how could you ensure that the LLM can extract EVERY relevant triplet in the questions, and if not, how this will impact the performance of KGARevion? To compare, LLM-based entity extraction is the core of [4], and the authors have employed specific strategies (i.e., multi-round prompting with follow-up questions) to avoid hallucination and ensure completeness.

**Explainability & trustworthiness.** Another concern I have regarding KGARevion is the lack of explainability. Since the veracity checking of triplets in KGARevion heavily relies on LLMs and KG embeddings, which are both black-boxes themselves, it would make it hard to justify why a provided answer is favored by KGARevion. This issue is especially critical in the medical context given its high-stakes nature. From my experience talking to clinicians in the UK, they would prefer a system that could faithfully justify the rational behind a decision rather than a high-performance black-box that you never know how it makes decisions. In this aspect, I think more conventionally yet transparent computational fact-checking methods like the Knowledge Linker (KL) or KG-Miner may offer more desirable behaviors and could at least be discussed in the paper. I do not think SOTA performance is the only measure for the success of a computational approach in the medical domain. Given these, I am wondering whether the authors have considered graph topology-based methods like KL or KM. How would they perform with KGARevion? Is there any relevant experiments data? And, how did the authors balance between performance and explainability?

---

> ### Public Comment · ~Mingyu_Huang1 · 2024-11-14
> **Additional points and references**
>
> **Readability.** As also mentioned by all four reviewers, I found this manuscript hard to follow (though I spent hours on it). Specifically, the motivation in the Introduction section is not logically clear (see, e.g., the "reliability" point above). More importantly, the methods section seems to be over complex (but not in the positive way). This makes it hard for me to know what exactly KGARevion performs at each step. This in turns hampers the reproducibility of the manuscript, as one cannot get enough technical details for fully replicating the approach after reading the manuscript (note that here by reproducibility I am more referring to the writing, not the source code). Therefore, it seems that the manuscript requires a thorough revision before it can be published in top venues like ICLR.
>
> **Computational cost and accessibility.** A final question I have is regarding the computational cost of the fine-tuning stage. Normally, this can be as least importance for a methodology paper in academia. However, since KGARevion uses KG to deal with medical QA---both are very sensitive to knowledge recency---one may then desire that the approach could be easily updated to leverage the newest KG at the moment (imagine that if KGARevion uses a KG proposed before 2019, then how could it accurately reason about questions regarding COVID-19?). However, I noticed that the authors mentioned in their Appendix, 4 H100 GPUs were used for fine-tuning the LLMs. This then seems to make the update of KGARevion very expensive. I am wondering about the exact cost of the fine-tuning stage (e.g., total GPU hours, memory consumption), and would this be accessible by labs with only one or two consumer-level GPU(s) (e.g., RTX 4090).
>
> I would appreciate it if the authors could kindly address my concerns (though I know they would be busy with responding to the reviewers).
>
> [1] H. Nori et al., Capabilities of GPT-4 on Medical Challenge Problems, arXiv (2023).
>
> [2] D. Veen et al., Adapted large language models can outperform medical experts in clinical text summarization, Nat. Med. (2024).
>
> [3] J. Yang et al., Poisoning medical knowledge using large language models, Nat. Mac. Intell. (2024).
>
> [4] M. Polak and D. Morgan, Extracting accurate materials data from research papers with conversational language models and prompt engineering, Nat. Comm. (2024).

---

> ### Author Response · Authors · 2024-11-29
> **Response to Public Comment**
>
> ### Evaluation (data leakage)
>
> Thank you for your comment. These are great points, which we addressed thoroughly through new experiments. Kindly see our response to **Review jrR1 on Weakness 1**.
>
>
> ### Reliability 1 (KG quality)
>
> Thank you for your comment.
>
> First, to clarify, we simply wanted to point out that existing RAG-based approaches are sensitive to the quality of retrieved information. There was no intention to criticize these methods. Second, thank you for sharing the recent work. However, the use of triplets to model and predict biomedical KGs is well-established, thoroughly validated, and widely used in research (for example, see following references), who provides a principled grounding for triplet-based representation of relational facts in KGARevion. Please see our response to **Review yzKB on Weakness 2** for additional information and discussion of these questions.
>
> *\[1\] Zifeng Wang, Zichen Wang, Balasubramaniam Srinivasan, Vassilis N. Ioannidis, Huzefa Rangwala and RISHITA ANUBHAI. BioBridge: Bridging Biomedical Foundation Models via Knowledge Graphs. In the Twelfth International Conference on Learning Representations, 2024\.*
> *\[2\] Cattaneo A, Martynec T, Bonner S, et al. Towards Linking Graph Topology to Model Performance for Biomedical Knowledge Graph Completion\[C\]//ICML'24 Workshop ML for Life and Material Science: From Theory to Industry Applications.*
> *\[3\] Pengcheng Jiang, Lang Cao, Cao Xiao, Parminder Bhatia, Jimeng Sun, and Jiawei Han. KG-FIT: Knowledge Graph Fine-Tuning Upon Open-World Knowledge. The Thirty-eighth Annual Conference on Neural Information Processing Systems, 2024\.*
> *\[4\] Yang, Junwei, et al. "Poisoning medical knowledge using large language models." Nature Machine Intelligence (2024): 1-13.*
> *\[5\]Teneva, Nedelina, and Estevam Hruschka. "Exploring the Properties and Structure of Real Knowledge Graphs across Scientific Disciplines." NeurIPS 2023 AI for Science Workshop. 2023\.*
>
> ### Reliability 2 (LLM for entity extraction)
>
> Thank you for your points. KGARevion is proposed to improve the accuracy of LLMs by incorporating structural knowledge from biomedical KGs, rather than addressing hallucinations in LLMs. The multi-round strategy is designed to ensure completeness. Existing LLMs (e.g., LLaMA3.1-8B) follows instructions well, effectively extracting entities contained in the input query. Regarding triplets, our goal is not to ensure that the LLM extracts every triplet related to the input question. Instead, we focus on analyzing the triplets generated by the LLM to understand its interpretation of the query. By verifying these triplets and removing any factually incorrect ones, we improve the LLM’s accuracy in responding to the query.
>
> ### Explainability & trustworthiness
>
> Thank you for your comment. Explainability and trustworthiness are indeed important topics for LLMs and represent separate standalone research areas in the LLM field. However, they are not the primary focus of KGARevion, as the KGARevion falls in the area of KG-based AI agents and thus we don’t claim any algorithmic/methodological advances that would relate to the areas of explainability. However, to address your question, KGARevion derives its final answer by considering all verified triplets, which collectively provide a reasoning path to support the outcome. Regarding fact-checking methods, we have also included related work in response to **Reviewer yzKB’s question Q5**.
>
> ### Readability
>
> Thank you for your suggestion. We have rewrite the section of the ‘Review’ action. Please refer to the response to **Reviewer jrR1 on Weakness 2.4**.
>
> ### Computational cost and accessibility
>
> We fine-tuned the LLM backbone (e.g., LLaMA3.1-8B) using 4 H100 GPUs, which took approximately 4-5 hours. The computational cost of KGARevion depends on the choice of the backbone LLM model, not its design or architecture. KGARevion is compatible with any LLMs and KGs, meaning that one can use KGARevion with smaller or larger language models as well as knowledge graphs of varying sizes, both of which we demonstrate in the paper (see Section 3.2.2 and 4.4 in the paper). We have implemented it with both LLaMA3-8B and LLaMA3.1-8B. The fine-tuning process is efficient and can also be performed using an RTX 4090 GPU.

---

### Author Response · Authors · 2024-11-24
**Global Response**

Thank you to all the reviewers for their thoughtful and insightful feedback\! We are pleased that reviewers are excited about the novel contributions of our work. Reviewers remark that “**KGARevion takes up the challenge of complex and difficult medical reasoning, making a relevant contribution**” \[XbfB\], “**KGARevion approach seems versatile, able to be used other base LLMs, KGs, and other domains in general**” \[jrR1\], “**KGARevion tackles an important problem and has strong evaluation and clarity**” \[yzKB\], and “**KGARevion holds significant importance in the field of medical question answering**” \[pkDu\] . We thank the reviewers for the strong praise of our work and contributions.

We now highlight a few important points raised by reviewers that warrant inclusion in the general response:

### Highlights of New Experiments

In response to reviewers' comments, we have run four additional experiments. We have included these experimental details and results in individual rebuttals. We now briefly describe these experiments:

* **New MedQA Dataset** \[jrR1\]**:** We included an external medical QA dataset (e.g., afrimedqa\_v2, released Sep. 12, 2024\) and tested KGARevion on this new dataset. This analysis is important because it is verifiably free from data leakage, and we find that KGARevion improves the accuracy by 4.7%, compared with LLaMA3.1-8B.
* **Additional Knowledge Graph** \[XbfB\]**:** We included an external KG (UMLS) and implemented the Review action in KGARevion with UMLS. We find that KGARevion is robust and compatible with any medical-related KGs since KGARevion (UMLS) improves the accuracy of LLaMA3.1-8B on all datasets.
* **Removed Triplets vs. Kept Triplets** \[yzKB\]**:** We conduct experiments to verify the number of triplets removed vs. kept by the Review action. We find that KGARevion is capable of improving the accuracy of backbone LLM (e.g., LLaMA3.1-8B) and enhancing the medical reasoning by removing those identified ‘False’ triplets.
* **Impact of irrelevant knowledge on KGARevion agent performance** \[yzKB\]: We add experiments to examine the impact of an out-domain KG (general-purpose KG with no medical domain-specialized knowledge) adopted in KGARevion. Even if KGARevion is used with an irrelevant or out-of-domain KG, KGARevion’s performance does not degrade substantially.

### Improved Writing for the 'Review' Action
In response to reviewers' comments \[jrR1,yzKB\], we revised sections on ‘Review’ action to enhance readability and clarity. In addition, we provided more details about the implementation of ‘Review’ and ‘Revise’ actions. The modifications can be found in the Appendix.

### Broader Context
In response to reviewers' comments \[yzKB\], we added related work on ‘fact-checking’ to make our discussion of background and related work more comprehensive. These additions can be found in the Related Work section.

### Enhanced Figure
We updated Figure 1 and Figure 2 to improve clarity and show the entire process reasoning process in the agent.

All updates are highlighted in blue in the revised PDF.

We thank all reviewers for their thoughtful commentary. We worked hard to improve our paper, and we sincerely hope the reviewers find our responses informative and helpful. If you feel the responses have not addressed your concerns to motivate increasing your score, we would love to hear what points of concern remain and how we can improve our work. Thank you again\!

---

### Meta-Review · Area_Chair_jabH · 2024-12-20

**Metareview:**

The authors introduce "KGARevion" for medical question answering. The method entails first generating structured triplets conditioned on a question and then verifying these against trusted knowledge bases. There was consensus this approach is novel and may have application beyond the immediate task. Moreover, the empirical results are promising (and concerns about leakage appear to have been adequately assuaged in discussion). Further, MEDDDX, which includes decoy options which are "semantically similar" to reference answers is a nice contribution unto itself.

I would appreciate discussion around the envisioned use of the model and associated risks. Even if KGARevion improves model factuality, it will still sometimes make mistakes; how should we think about using such a method in practice, especially in the absence of transparency (an issue raised by yzKB and the public commenter)? Some discussion of such issues and risks of using medical QA systems in practice should really be included in the work.

**Additional Comments On Reviewer Discussion:**

Reviewers raised critical points about potential data leakage, but the authors appear to have addressed these satisfactorily in the discussion period.

---

### Decision · Program_Chairs · 2025-01-22

Accept (Poster)